# Respiratory and cardiac interoceptive sensitivity in the first two years of life

Markus R Tünte[1,2]*, Stefanie Hoehl[1], Moritz Wunderwald[1], Johannes Bullinger[1,3], Asena Boyadziheva[1], Lara Maister[4], Birgit Elsner[5], Manos Tsakiris[6], Ezgi Kayhan[5]

[1]University of Vienna, Faculty of Psychology, Department of Developmental and Educational Psychology, Vienna, Austria; [2]Vienna Doctoral School Cognition, Behavior and Neuroscience, University of Vienna, Vienna, Austria; [3]Ludwig-Maximilians-Universität München, Munich, Germany; [4]School of Human and Behavioural Sciences, College of Human Sciences, Prifysgol Bangor University, Gwynedd, United Kingdom; [5]Department of Developmental Psychology, University of Potsdam, Potsdam, Germany; [6]Department of Psychology, Royal Holloway University of London, London, United Kingdom

*For correspondence:
markus.tuente@univie.ac.at

Competing interest: The authors declare that no competing interests exist.

## eLife Assessment

This study presents **important** findings on the early development of cardiac and respiratory interoceptive sensitivity based on an investigation of infants aged 3, 9 and 18 months and on extensive statistical analyses. The evidence supporting the conclusions are **convincing** although the research faced technical challenges that limited part of the findings interpretation. This study will be of significant interest to developmental psychologists and neuroscientists working on interoception and its influence on socio-cognitive development.

**Abstract** Several recent theoretical accounts have posited that interoception, the perception of internal bodily signals, plays a vital role in early human development. Yet, empirical evidence of cardiac interoceptive sensitivity in infants to date has been mixed. Furthermore, existing evidence does not go beyond the perception of cardiac signals and focuses only on the age of 5–7 mo, limiting the generalizability of the results. Here, we used a modified version of the cardiac interoceptive sensitivity paradigm introduced by Maister et al., 2017 in 3-, 9-, and 18-mo-old infants using cross-sectional and longitudinal approaches. Going beyond, we introduce a novel experimental paradigm, namely the iBREATH, to investigate respiratory interoceptive sensitivity in infants. Overall, for cardiac interoceptive sensitivity (*total n*=135) we find rather stable evidence across ages with infants on average preferring stimuli presented synchronously to their heartbeat. For respiratory interoceptive sensitivity (*total n*=120) our results show a similar pattern in the first year of life, but not at 18 mo. We did not observe a strong relationship between cardiac and respiratory interoceptive sensitivity at 3 and 9 mo but found some evidence for a relationship at 18 mo. We validated our results using specification curve- and mega-analytic approaches. By examining early cardiac and respiratory interoceptive processing, we provide evidence that infants are sensitive to their interoceptive signals.

## Introduction

Bodily functions such as heartbeat and respiration are vital to the survival of living beings. The perception of signals arising from the body such as heartbeat, respiration, and hunger is called interoception (*Craig, 2002*). Individuals differ with regard to their interoceptive sensitivity, the degree to which they

perceive their own bodily signals (*Critchley and Harrison, 2013*). Interoceptive sensitivity is related to human experience and behavior, such as the perception of emotions, mental health, and social cognition (*Khalsa et al., 2018*). Furthermore, several recent theoretical accounts have highlighted that interoceptive sensitivity plays a vital role in early development in infancy, such as the development of the self and early social abilities (*Filippetti, 2021*; *Fotopoulou and Tsakiris, 2017*; *Musculus et al., 2021*). As infants are born with limited ability to self-regulate bodily states giving rise to interoceptive sensations such as hunger, they rely on interactions with their primary caregiver for co-regulation. These interactions in turn play an important role in shaping early development in infancy. Despite these theoretical frameworks, we have little knowledge about infants' sensitivity to their interoceptive signals. Recently, the first paradigm to assess cardiac interoceptive sensitivity in infancy was introduced (*Maister et al., 2017*). In the present study, we aim to replicate the experimental paradigm introduced by *Maister et al., 2017* on cardiac interoceptive sensitivity in infants. Furthermore, we aim at tracking the development of interoceptive sensitivity and related individual differences across the infancy period. Going beyond cardiac perception, we introduce a novel approach to measure respiratory interoceptive sensitivity in infants.

Most empirical investigations of interoceptive processing have focused on cardiac interoception (*Khalsa et al., 2018*). In adults, a large body of research has investigated cardiac interoceptive perception using paradigms in which participants are asked to count or detect their own heartbeat (*Brener and Ring, 2016*; *Schandry, 1981*). Using modified versions of the tasks for adult participants, studies with children have shown that stable and adult-like interoceptive skills can be measured already at 4–6 y of age (*Schaan et al., 2019*).

In contrast to the existing evidence on cardiac interoception in children, we know little about whether infants perceive their interoceptive signals. The first published empirical evidence on interoceptive sensitivity used an eye-tracking paradigm, namely the iBEATs task, in which 5-mo-old infants observed images on the screen such as clouds and stars that bounced either synchronously or asynchronously to the infant's heartbeat. *Maister et al., 2017* found that infants on average looked longer at stimuli that moved asynchronously to their heartbeat as compared to stimuli moving synchronously. Furthermore, infants' cardiac interoceptive sensitivity scores were correlated with their heartbeat-evoked potentials (HEPs), a neural marker of interoceptive processing (*Coll et al., 2021*). This study provided the first evidence that already at 5 mo of age infants show sensitivity to their own cardiac signals. Furthermore, this approach has also successfully been replicated with rhesus monkeys (*Charbonneau et al., 2022*) and a recent study using an adapted experimental paradigm in 6-mo-old infants has found similar results (*Imafuku et al., 2023*).

Recently, however, no evidence of cardiac interoceptive sensitivity in 5- to 7-mo-old infants was reported (*Weijs et al., 2023*). Despite some methodological differences, all studies used very similar experimental paradigms in which infants were presented with stimuli oscillating either synchronously or asynchronously to their heartbeat. It is unclear whether the null findings reported by *Weijs et al., 2023* indicate that infants at this age do not show cardiac interoceptive sensitivity or whether methodological differences, such as the measurement method, outlier rejection criteria, and statistical power, might explain divergent results. In any case, the findings of the study by *Weijs et al., 2023* highlight the importance of replicating the iBEATs paradigm (*Maister et al., 2017*) to advance the understanding of interoception in infants.

To gain a more comprehensive understanding of interoceptive processing in infancy, it is crucial to investigate other interoceptive modalities. Especially, as different interoceptive modalities are not necessarily related and might have different functions or underlying neural signatures (*Allen et al., 2023*; *Garfinkel et al., 2016*; *Khalsa et al., 2018*). One interoceptive signal that is closely related to cardiac processes is respiration. In fact, heartbeat and respiration are linked functionally and anatomically (*Draghici and Taylor, 2016*; *Garcia et al., 2013*).

In recent years, an increasing number of publications have focused on the perception of respiration in adults (*Weng et al., 2021*). Experimental paradigms that measure sensitivity to resistance in breathing, for instance, have highlighted the connection between respiratory interoception and emotional states such as anxiety (*Harrison et al., 2021a*; *Nikolova et al., 2022*; *Harrison et al., 2021b*). The neural network which links respiratory perception with emotional and cognitive processes has become a matter of scientific interest (*Allen et al., 2023*; *Kluger et al., 2021*). The emerging literature shows that breathing constitutes a fundamental process with functional significance for

self-regulation (*Boyadzhieva and Kayhan, 2021*; *Heck et al., 2016*). In children, it has been shown that sensitivity to respiratory signals can be observed from at least 10 y of age onward (*Nicholson et al., 2019*). Given the relevance of respiration for self-regulation and social interaction, it is important to map out respiratory interoceptive sensitivity in infancy, a period that is especially relevant for development of self-related perception (*Van Puyvelde et al., 2019*; *Weng et al., 2021*).

Regarding the relationship between cardiac and respiratory interoception, empirical results have painted a mixed picture. In children, it has been reported that respiratory interoception is not correlated to cardiac interoception (*Nicholson et al., 2019*). In adults, it has been found that while accuracy in interoceptive domains is not related across cardiac and respiratory perception, metacognitive awareness for both modalities shows a significant relation (*Garfinkel et al., 2016*). In early infancy, the relationship between different interoceptive modalities has not yet been investigated. Therefore, it is currently unclear whether sensitivity to different interoceptive modalities emerges at the same time, and in a similar manner.

In the present study, we aim to fill the knowledge gap on interoceptive sensitivity in early infancy by reporting results from two studies investigating cardiac and respiratory interoceptive sensitivity in 3-, 9- and 18- mo-old infants. We investigated the age group of 3 mo to provide evidence on the early emergence of interoceptive sensitivity, as 3 mo is the earliest at which eye tracking paradigms can be reliably applied. We chose the groups of 9 and 18 mo as these ages mark important milestones in the development of abilities related to interoception, such as self-perception (*Filippetti, 2021*; *Fotopoulou and Tsakiris, 2017*; *Musculus et al., 2021*) and social cognition (*Carpenter et al., 1998*; *Repacholi and Gopnik, 1997*). For instance, around 18 mo of age explicit mirror self-recognition can be observed (*Amsterdam, 1972*). Moreover, around 9 mo of age, the ability to show joint attention drastically matures (*Carpenter et al., 1998*).

To measure cardiac interoceptive sensitivity, we used a modified version of the iBEATs paradigm originally developed by *Maister et al., 2017*, as it is the only task to measure cardiac interoceptive sensitivity in infants to date. Moreover, we developed a novel experimental paradigm that follows the logic of the iBEATs to investigate respiratory interoceptive sensitivity in infants: the iBREATH task. In contrast to previous tasks, the present study used a fixed-experimental paradigm due to a technical error (i.e. the presentation sequence was consistent across participants; see Appendix 1). In a first step, we conducted a study with a longitudinal design investigating 9- and 18-mo-old infants. We then

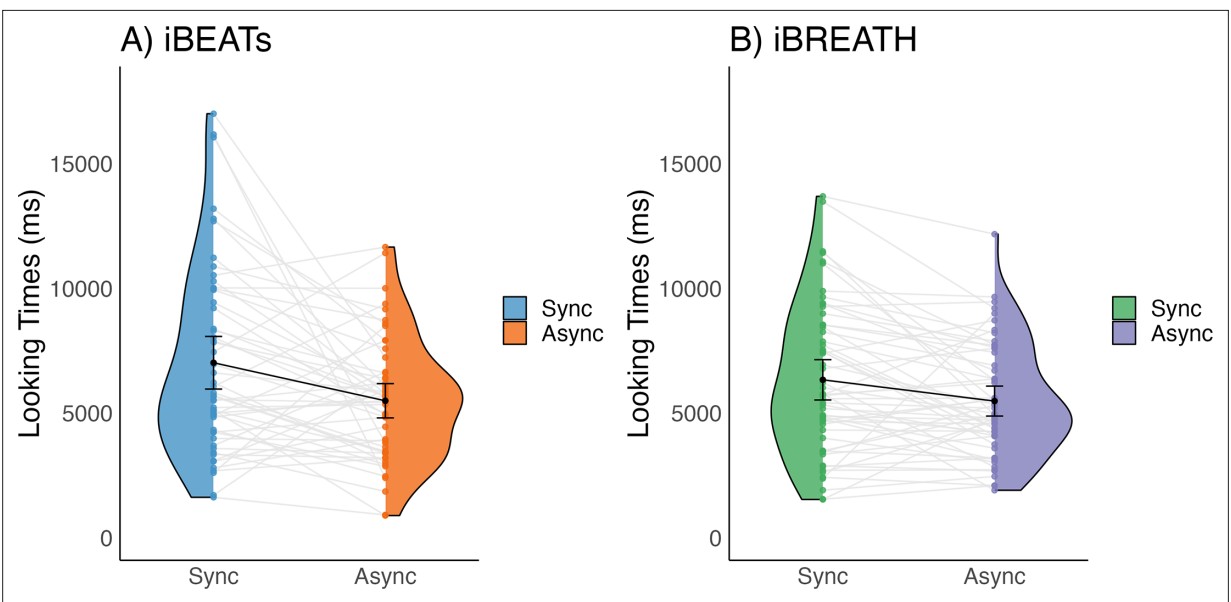

**Figure 1.** Looking times for (**A**) iBEATs and (**B**) iBREATH in 9-mo infants. Looking times for the (**A**) iBEATs (paired t-test, N = 52, t=–2.96, p=0.005) and (**B**) iBREATH (paired t-test, N = 56, t=–2.80, p=0.007) tasks. In both tasks, 9-mo-old infants looked significantly longer at stimuli presented synchronously to their own physiological signals. Black dots refer to the group mean. Black bars refer to the standard error of the mean. Gray lines and colorful dots refer to individual mean-looking times per condition and infant. In (**A**) blue represents the synchronous condition and orange is the asynchronous, while in (**B**) green represents the synchronous condition, and purple is the asynchronous.

replicated the experimental paradigms in a separate sample of 3-mo-old infants. Our initial prediction concerning the 9- and 18-mo-old sample were threefold: (1) For both tasks and in both age groups (9 and 18 mo), we expected to find the same preference in looking behavior as reported in *Maister et al., 2017*, that is, longer looking times to asynchronous trials as an indication of infants' detection of incongruency of the visual stimulus and their interoceptive signals. (2) We expected to find a positive correlation between performance in the cardiac and respiratory interoception tasks given the conceptual proximity between both tasks. Adult cardiac and respiratory interoception paradigms typically use two conceptually different paradigms. Thus, null results in the adult literature might be due to the unique characteristics of those paradigms. Last, (3) we predicted an increase in individual interoceptive sensitivity from 9 to 18 mo of age, as increased interoceptive sensitivity might be associated with the development of self-recognition and socio-cognitive skills. We tentatively predicted that 3-mo-olds would already differentiate between visual displays that move in synchrony vs. asynchrony to their own cardiac and respiratory signals.

## Results

### Confirmatory analyses: 9-mo-old infants

First, we investigated whether 9-mo-old infants displayed sensitivity to their cardiac and respiratory signals. Following our preregistered analysis plan (https://aspredicted.org/QP9_6FP) we computed paired t-tests to compare mean looking times between synchronous and asynchronous conditions for the iBEATs and the iBREATH. We found that for both tasks 9-mo-old infants displayed a preference for stimuli presented synchronously with their own heartbeat (*Figure 1A*, N=52, $M_{synch}$ = 7020.62 ms $SD_{synch}$ = 3790.00 ms, $M_{asynch}$ = 5496.7 ms, $SD_{asynch}$ = 2469.34 ms, $t$=–2.96, p=0.005, Cohens $d$=0.48) and respiration (*Figure 1B*, N=56, $M_{synch}$ = 6336.21 ms, $SD_{synch}$ = 3017.37 ms, $M_{asynch}$ = 5483.77 ms, $SD_{asynch}$ = 2244.84 ms, $t$=–2.80, p=0.007, Cohens $d$=0.32). These results on the one hand replicate the approach by *Maister et al., 2017* in an older age group showing that infants are sensitive to their cardiac signals. Going beyond, using a novel paradigm, we further provide the first evidence that infants are also sensitive to their respiratory signals. Notably, mean preferences were switched compared to our expectations and the results of *Maister et al., 2017* who reported a mean preference for stimuli presented asynchronously to the infants' heartbeats.

### Interoceptive sensitivity at 18 mo

Next, we followed up with the same infants at 18 mo. Unfortunately, as the study was conducted during the Covid-19 pandemic, we had a large number of dropouts for the longitudinal follow-up. We conducted paired t-tests comparing looking times to synchronous and asynchronous stimuli at 18 mo following our approach of the 9-mo-old sample for iBEATs (N=28, $M_{synch}$ = 6924.16 ms $SD_{synch}$ = 3833.01 ms, $M_{asynch}$ = 6427.68 ms, $SD_{asynch}$ = 3801.02 ms, t(27) = –0.75, p=.461, $d$=0.13) and iBREATH (N=30, $M_{synch}$ = 3612.95 ms $SD_{synch}$ = 1879.02 ms, $M_{asynch}$ = 4098.71 ms, $SD_{asynch}$ = 2074.11 ms, t(29) = 1.09, p=.283, $d$=–0.25) which did not indicate a significant mean preference. However, a non-significant result does not provide evidence for the absence of an effect (*Lakens et al., 2018*). Therefore, we conducted two equivalence tests using the effect size reported by *Maister et al., 2017*, $d$=0.4) as equivalence bounds. Equivalence tests facilitate the interpretation of non-significant results by investigating whether a given confidence interval is too wide to discriminate between expected effect (=the equivalence bounds) and null effect, or whether one can rule out an effect at least as strong as we expected. The results of the equivalence tests indicate that we do not find conclusive evidence in favor of or against a mean preference effect in our 18-mo-old sample for both the iBEATs (t(27) = 0.71, p=0.242) or the iBREATH ((t(29) = 1.10, p=0.141), potentially due to the small sample size.

When inspecting results from our analysis approach (e.g. *Figure 1*), as well as previous results (*Maister et al., 2017*; *Weijs et al., 2023*) it becomes evident that there are large individual differences in preferences (e.g. some infants prefer synchronous, some asynchronous trials). Thus, sample size might be an important factor in detecting a mean preference effect. To gain additional insights into the interplay of sample size and variability due to the large individual differences we conducted simulations which are reported in Appendix 2. Overall, results from the simulation indicate that sample sizes of around 30 infants might be too small to reliably detect a mean preference effect in the version of the iBEATs task used here.

## Interoceptive sensitivity at 3 mo

Initially, the present project was planned as a longitudinal approach spanning 3-, 9-, and 18 mo. However, difficulties in recruiting very young infants due to the Covid-19 pandemic precluded us from starting the longitudinal assessment with 3-mo-old infants. Still, we decided to test the iBEATs and iBREATH tasks in an additional 3-mo-old sample once recruitment was possible again (pre-registration: https://aspredicted.org/44L_QKH). Data for this sample was collected after analysis of the 9- and 18-mo-old data. Using our preregistered analysis approach, we found evidence for a group mean preference for synchronous stimuli in the iBEATs (paired Bayesian t-test; BF = 2.02, mean difference: 793.95 ms, 95% CI [108.63, 1388.69], N=53, $M_{synch}$ = 6131.13 ms $SD_{synch}$ = 5129.93 ms, $M_{asynch}$ = 5337.17 ms, $SD_{asynch}$ = 5044.83 ms) but not in the iBREATH task (paired Bayesian t-test; BF = 0.23, mean difference: 502.21 ms, 95% CI [–701.49, 1600.86], N=40, $M_{synch}$ = 7881.72 ms $SD_{synch}$ = 7641.50 ms, $M_{asynch}$ = 7379.50ms, $SD_{asynch}$ = 7220.50 ms) at 3 mo of age. Due to the absence of evidence for the iBREATH task we conducted a test for practical equivalence similar to our approach for the 18-mo-old's data (*Harms and Lakens, 2018*). We used the effect size of the iBREATH task at 9 mo to investigate whether we can rule out an effect at least as strong as that. Results indicated that we cannot distinguish between absence or presence of an effect at least as strong as it was present in the 9-mo-old's iBREATH sample (95% HDIs = [–711.41, 1606.80], region of practical equivalence: 77.08%). Reasons for the non-significant result might include the smaller sample size for the iBREATH at 3 mo (N=40) compared to the iBEATs (N=53), combined with a reduced signal-to-noise ratio for eye-tracking tasks in 3-mo-olds compared to older infants, in general. In sum, we replicate the results of our 9-mo-old sample for the cardiac domain in 3-mo-old infants, while finding inconclusive evidence regarding the respiratory domain.

## Interoceptive sensitivity in the first two years of life – a MEGA analytic approach

So far, we have presented results on cardiac and respiratory interoceptive sensitivity spanning three age groups in the first two years of life. We find some evidence that infants prefer stimuli presented synchronously with their respective physiological signals. However, we also find some inconclusive

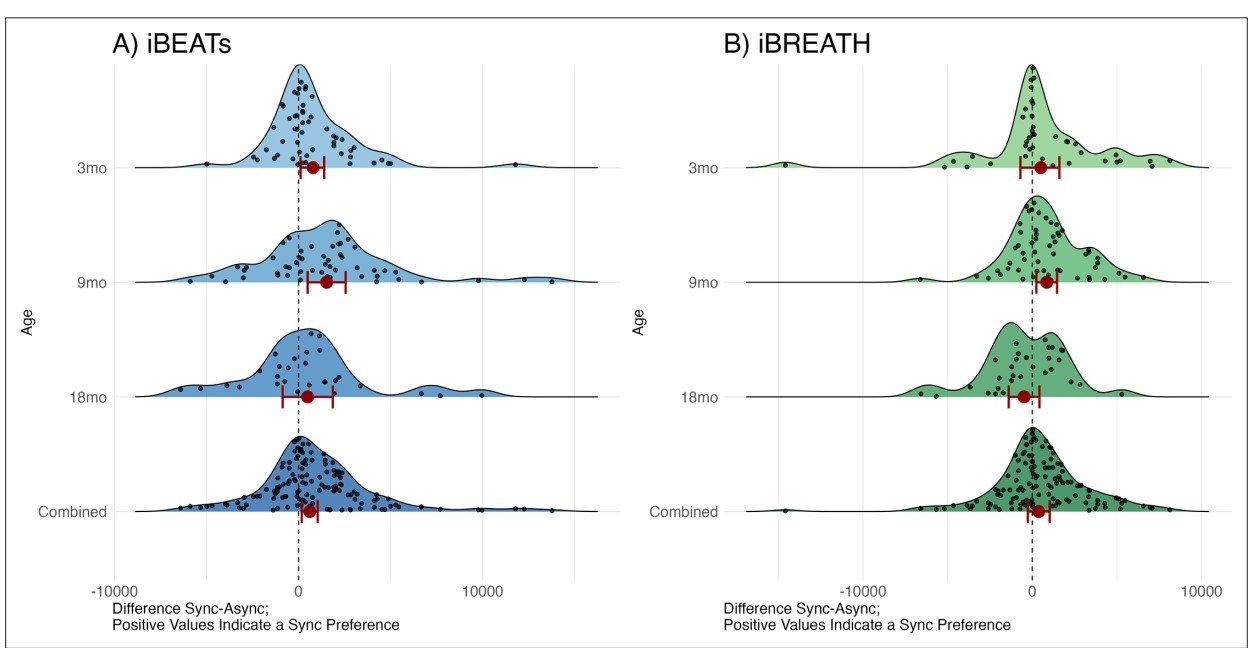

**Figure 2.** Results from the MEGA analysis for (**A**) iBEATs and (**B**) iBREATH. Results from the MEGA analysis for (**A**) iBEATs (combined sample N = 135) and (**B**) iBREATH (combined sample N = 120). Plot of difference scores computed as mean synchronous looking times minus mean asynchronous looking times per individual for each age group, as well as the combined sample. Red dots refer to mean effects for the respective analysis as described above, red bars refer to 95% confidence/credible intervals. Dashed line indicates a difference of 0. For 3, 9, and 18 mo age groups our preregistered analysis is plotted. For the combined sample we computed a linear mixed model using lme4 for visualization purposes as results from a mixed model with a beta error distribution cannot easily be transformed back to the original scale.

**Table 1.** Interactions between condition and age for the iBEATs MEGA-analysis.

| Term | Estimate | SE | z-value | p-value |
|---|---|---|---|---|
| Condition * age (3- vs 9 mo) | 0.00 | 0.11 | 0.02 | 0.982 |
| Condition * age (3- vs 18 mo) | 0.14 | 0.13 | 1.08 | 0.283 |
| Condition * age (9- vs 18 mo) | 0.14 | 0.12 | 1.12 | 0.264 |

Results for the mixed model using a beta error distribution. Here only the interaction between condition and age are reported for all combinations. Detailed results can be found in Appendix 3.

evidence, such that we cannot distinguish between a null-finding and a significant effect. This might be potentially due to a small number of observations in some of our samples, as indicated by equivalence tests and data simulation. So far, we have investigated all age groups separately, building up on our preregistration, and the assumption that results might be different for age groups. An alternative approach that might help in adjusting for sample size issues is to pool together all age groups using an explorative MEGA-analytic approach (*Koile and Cristia, 2021*). Such an approach might give us the statistical power needed to make claims about absence or presence of a cohesive effect in the first two years of age, i.e., whether the mean effect across age groups supports the conclusion of a shared effect.

We computed two mixed models with looking time as outcome, condition and age-group/experiment, as well as their interaction, as fixed effects, and participant as a random effect using the R-package 'glmmTMB' (*Brooks et al., 2017*) utilizing a beta error distribution for the iBEATs (*Figure 2A* bottom, *Table 1*) and the iBREATH (*Figure 2B* bottom, *Table 2*), respectively. This approach allowed us to include 135 observations for the iBEATs from 125 infants, and 120 observations for the iBREATH from 107 infants. The sample size differs slightly from our preregistered approach given that we used the same preprocessing approach for the MEGA-analysis for all samples.

First, we compared each model with a null model missing the condition term. For the iBEATs we find that the full model is statistically significant from the null model, suggesting a better fit (p=0.012). For the iBREATH we do not find a statistically significant better fit for the full compared to the null model (p=0.091). Still, the Bayesian information criterion (BIC), which can be interpreted similar to a Bayes Factor (*Burnham and Anderson, 2004*), related to this comparison is 15.1 smaller for the full (BIC$_{full}$ = −1574.5), compared to the null model (BIC$_{null}$ = −1559.4), giving some evidence for a better fit for the full model.

Next, we inspected the model output. For both models, we did not find a significant interaction between age and condition indicating that the effect of condition on age group does not significantly vary between age groups (iBEATs: *Table 1*, iBREATH: *Table 2*). Next, we computed post hoc comparisons using estimated marginal means from the MEGA-analysis across all age groups to investigate whether we find indications for a similar effect across ages. For the iBEATs, we found a significant main effect of condition on looking time in the combined sample indicating that infants show longer looking times for stimuli presented synchronously with their heartbeat over all ages (OR = 1.13, 95% CI [1.03, 1.25], t(1769)=2.541, p=0.011). In contrast, for the iBREATH, we did not find a significant effect of condition on looking time over all ages (OR = 1.07, 95% CI [0.96, 1.20], t(1284)=1.192, p=0.234). Interestingly, we find that all samples and tasks apart from the 18-mo-old iBREATH sample show a numerical preference for synchronous stimuli. In reporting these results

**Table 2.** Interactions between condition and age for the iBREATH MEGA-analysis.

| Term | Estimate | SE | z-value | p-value |
|---|---|---|---|---|
| Condition * age (3- vs 9 mo) | 0.02 | 0.12 | 0.17 | 0.864 |
| Condition * age (3- vs 18 mo) | 0.23 | 0.16 | 1.50 | 0.134 |
| Condition * age (9- vs 18 mo) | 0.21 | 0.15 | 1.43 | 0.154 |

Results for the mixed model using a beta error distribution. Here only the interaction between condition and age are reported for all combinations. Detailed results can be found in Appendix 3.

**Table 3.** Effects of iBEATs on iBREATH for all age groups, as well as for interactions between iBEATs and age.

| Term | Estimate | SE | z-value | p-value |
|---|---|---|---|---|
| iBEATs (3 mo) | −1.83 | 0.97 | −1.89 | 0.059 |
| iBEATs (9 mo) | −1.16 | 0.90 | −1.30 | 0.192 |
| iBEATs (18 mo) | 1.30 | 1.02 | 1.27 | 0.204 |
| iBEATs * age (3- vs 9 mo) | −0.15 | 0.34 | −0.42 | 0.674 |
| iBEATs * age (3- vs 18 mo) | 3.13 | 1.41 | 2.22 | 0.027 |
| iBEATs * age (9- vs 18 mo) | 2.45 | 1.36 | 1.81 | 0.070 |

Results for the beta regression of iBEATs scores on iBREATH scores. Detailed results can be found in Appendix 3.

we focus on whether we found evidence for interactions between age groups, and whether we found evidence for a general effect across age groups. In-depth results and tables can be found in Appendix 3.

To sum up, regarding cardiac interoceptive sensitivity, results from the MEGA analysis support the notion that, across all age groups tested here, infants on average prefer stimuli presented synchronously with their own heartbeat. Regarding respiratory interoceptive sensitivity, we only found evidence in our 9-mo-old sample, but not in the 3- and 18- mo-olds, or the MEGA analysis. However, this latter result might be driven by the 18-mo-old iBREATH sample .

## The relationship between cardiac and respiratory interoceptive sensitivity

Next, we investigated the relationship between cardiac and respiratory interoceptive sensitivity. First, we computed absolute proportional scores as individual difference scores for the iBEATs and the iBREATH following previous approaches and our preregistration. These scores range from 0 to 1, and a higher score indicates a stronger preference for either synchronous or asynchronous stimuli in the iBEATs or iBREATH, respectively. However, a difference score does not indicate the direction of the preference (synchronous or asynchronous). The reasoning behind the use of absolute proportional scores is that, in principle, both a preference for synchronous and for asynchronous stimuli indicates that the participant identified a (bodily) signal from noise. Importantly, all studies using iBEATs like paradigms in infants so far have used absolute proportional scores to investigate individual differences (*Maister et al., 2017*; *Weijs et al., 2023*). Furthermore, visual inspection of the individual preferences in both paradigms (gray lines, *Figure 1*) reveals that, although the group mean difference displays a preference for the synchronous stimuli, in fact, looking preferences for both synchronous and asynchronous stimuli can be observed on an individual level.

We used a mega-analytic approach, pooling together data from all age groups, to investigate the relationship between both tasks. We fitted a mixed model using a beta-error distribution with the iBREATH scores as outcome, the iBEATs, age, and their interaction as factors, and participant as a random factor (*Table 3*, for detailed results see Appendix 3). We did not find a strong relationship between cardiac and respiratory interoceptive sensitivity across all ages (*Figure 3*, N=84), mirroring previous results in adults and children (*Garfinkel et al., 2016*; *Nicholson et al., 2019*). However, we found a significant interaction between the iBEATs scores and age, specifically comparing the 3- and 18-mo-old groups ($\beta$=3.13, SE = 1.41, p=0.027). This interaction indicates that the relationship between iBEATs and iBREATH scores changes between 3 and 18 mo of age.

To follow up the interaction, we conducted a pairwise comparison which indicated that for the effect of iBEATs scores on the iBREATH scores there was a significant difference between 9- and 18 mo of age ($\beta$=−0.60, SE = 0.24, p=0.043), while there were no significant differences between 3- and 18 mo ($\beta$=−0.60, SE = 0.25, p=0.055) or the 3- and 9-mo-olds ($\beta$=0.00, SE = 0.23, p=0.999). Still, coefficients indicate a similar strength and direction of the comparison between 9 and 18 mo as well as 3 and 18 mo.

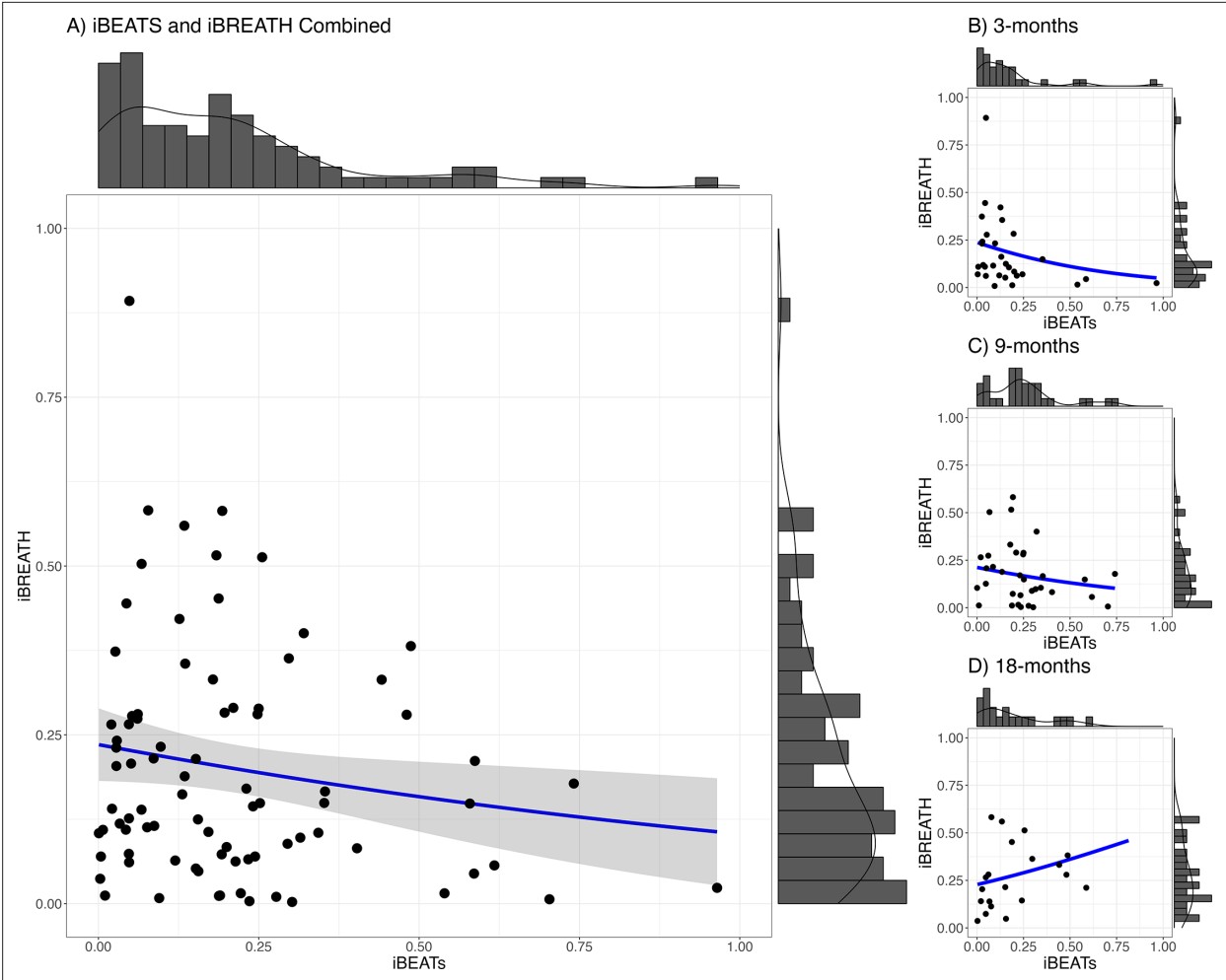

**Figure 3.** Relationship between iBEATs and iBREATH using a combined sample. Histogram with a plotted line for individual performance on iBEATs and iBREATH using a beta regressionN = 84. Following *Maister et al., 2017*, individual difference scores were computed as the proportion of the absolute difference between synchronous and asynchronous trials.

## Developmental changes in interoceptive sensitivity

Next, we aimed at further investigating whether there are developmental changes in interoceptive sensitivity in the first two years of life. Initially, following our preregistration (https://aspredicted.org/GMB_XCW), we conducted a longitudinal analysis using the infants that participated both at 9 and 18 mo of age. Unfortunately, as described earlier, due to the study being conducted during the Covid-19 pandemic, only a subsample of infants could be re-invited to the lab in the targeted age range and contributed data suitable for longitudinal analyses. Comparing the absolute individual difference scores between both age groups revealed no evidence for a strong change in cardiac (paired Bayesian t-test; BF = 0.26, N=20) or respiratory (paired Bayesian t-test; BF = 0.33, N=19) interoceptive sensitivity, indicating that absolute individual difference scores in both domains do not change substantially from 9 to 18 mo of age. Notably, a region of practical equivalence follow-up analysis indicates that we cannot rule out an effect at least as strong as a change of .1 for the absolute proportional scores (iBEATs: ROPE [–0.10, 0.10], 97.53% inside ROPE, iBREATH: ROPE [–0.10, 0.10], 97.76% inside ROPE, 95% HDI [–0.11, 0.05]). Furthermore, in an exploratory analysis, we computed Spearman correlations between timepoints. We did not find evidence for the iBEATs ($r(18)$ = 0.236, p=0.315) and the iBREATH ($r(17)$ = .195, p=0.423) that individual difference scores correlate strongly between timepoints.

To increase the number of observations, and statistical power, we conducted an exploratory MEGA-analytic follow-up in which we included all infants, not only those that contributed usable data to both

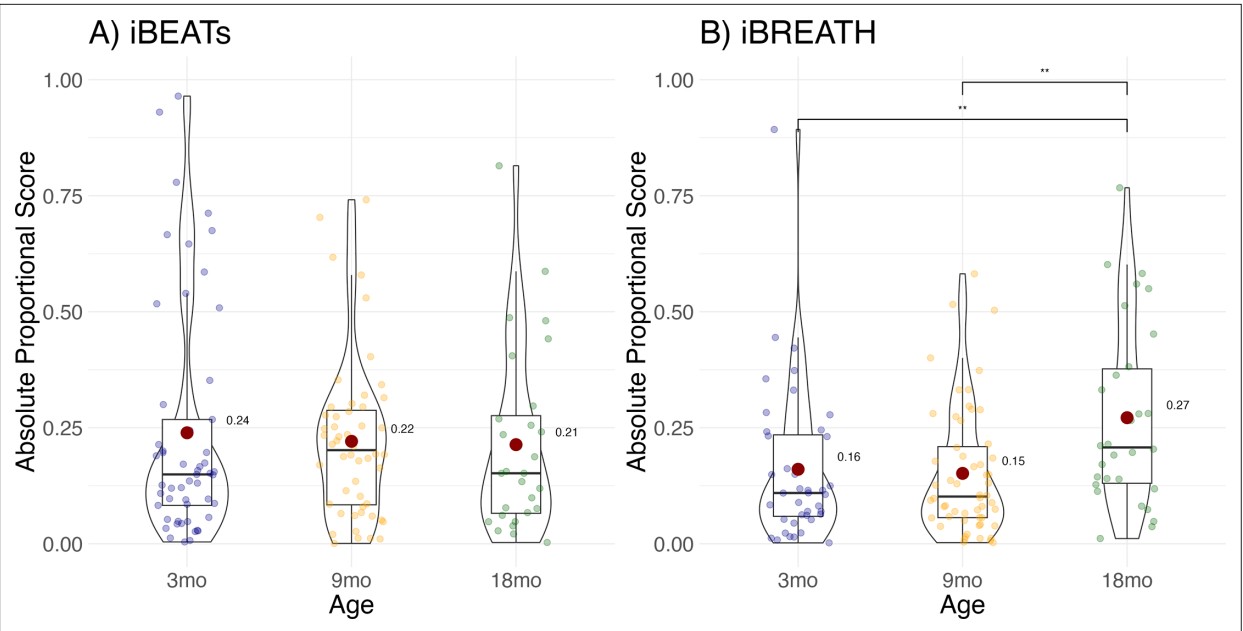

**Figure 4.** Exploratory analysis for age effect. Absolute proportional scores for (**A**) iBEATs and (**B**) iBREATH plotted for each age group. Red dots refer to group means, and colorful dots to individual means, ** refers to a significant result (p<0.01).

time points. Results showed that individual difference scores increased significantly for the iBREATH (*Figure 4B*, *Table 4*) in the 18-mo-olds compared to the 3-mo-olds (OR = 0.544, SE = 0.12, p=0.014), and the 9-mo-olds (OR = 0.525, SE = 0.12, p=0.004), but not for the iBEATs (*Figure 4A*, *Table 5*) indicating that respiratory, but not cardiac, interoceptive sensitivity increases at 18 mo of age.

## Specification curve analysis

Notably, apart from the 18-mo-old iBREATH sample, we found that (numerical) mean group preferences indicated a preference to stimuli presented synchronously with the respective bodily signal. Thus, mean group preferences were switched compared to our initial expectation and the original study by *Maister et al., 2017* who found a mean group preference for stimuli presented asynchronously to the infant's heartbeat. In addition, other studies have failed to find evidence for cardiac interoceptive sensitivity in infants (*Weijs et al., 2023*). Furthermore, a wide range of analytical choices have been reported in approaches on cardiac interoception in infants (*Maister et al., 2017*; *Weijs et al., 2023*) and nonhuman primates *Charbonneau et al., 2022* to date.

Therefore, it is important to further describe and validate our results. Using a specification curve analysis (*Simonsohn et al., 2020*), it is possible to map out the space of theoretically justified analysis strategies on a given dataset. Thus, it is possible to investigate whether analytical choices, such as differences in exclusion criteria and physical signal preprocessing, impact the results. Importantly, this method allows us to rule out that a group mean preference for synchronous stimuli is due to specific analytical choices of our preregistered analysis or whether a range of different analysis paths come to the same conclusion.

**Table 4.** Change in absolute proportional scores across age groups for the iBREATH.

| Term | Estimate | SE | z-value | p-value |
|---|---|---|---|---|
| Intercept | −1.61 | 0.15 | 10.55 | <0001 |
| 9 mo | −0.04 | 0.19 | −0.18 | 0.853 |
| 18 mo | 0.61 | 0.21 | 2.86 | 0.004 |

Results for the mixed model using a beta error distribution. Results are in comparison to the synchronous condition, and 3 mo age group.

**Table 5.** Change in absolute proportional scores across age groups for the iBEATs.

| Term | Estimate | SE | z-value | p-value |
| --- | --- | --- | --- | --- |
| Intercept | −1.11 | 0.14 | −7.91 | <0.001 |
| 9 mo | −0.04 | 0.19 | −0.22 | 0.826 |
| 18 mo | −0.09 | 0.23 | −0.41 | 0.684 |

Results for the mixed model using a beta error distribution. Results are in comparison to the synchronous condition, and 3 mo age group.

We ran a specification curve analysis following the approach outlined by *Simonsohn et al., 2020*. We used our 9-mo-olds dataset as input dataset, as it shows the clearest evidence for infant interoceptive sensitivity (i.e. better data quality compared to the 3 mo sample, and larger sample size compared the 18 mo sample). First, we identified theoretically justified analysis paths applicable to the present dataset by comparing the approaches presented in *Weijs et al., 2023*, *Maister et al., 2017* and *Charbonneau et al., 2022*. As the first step, we focused on the iBEATs, as already three different research groups have published experiments similar to the iBEATs and thus, a number of different specifications could be extracted from the literature (e.g. regarding physiological signal processing, in/exclusion criteria for infants and number of trials, or statistical analysis; for a full list see Appendix 4). Combining all possible ways of analyzing the present dataset gave a number of 1024 possible analyses which we subsequently ran (*Figure 5*). Next, we ran a specification curve for the iBREATH data of our 9-mo-old sample by extracting and adapting analytical decisions we used for the iBEATs, which resulted in 1536 possible analyses (*Figure 5B*).

Our results indicated that for the iBEATs almost half (44.73%) of all analytical paths led to a significant result (p<0.05), while for the iBREATH 17.51% of all analytical paths came to such a conclusion. Almost all specifications indicated a mean group preference for synchronous trials (43.16%, *Figure 5*, blue color). Interestingly, however, there were also a handful of specifications for the iBEATs (N=16, 1.6%) that would have resulted in a mean group preference for asynchronous trials (*Figure 2A*, red color). In sum, these results can be seen as a validation of our preregistered analytical approach described above, as they highlighted that a mean group preference for synchronous trials is not dependent on the combination or interaction of specific analytical choices. Still, given that 1.6% of

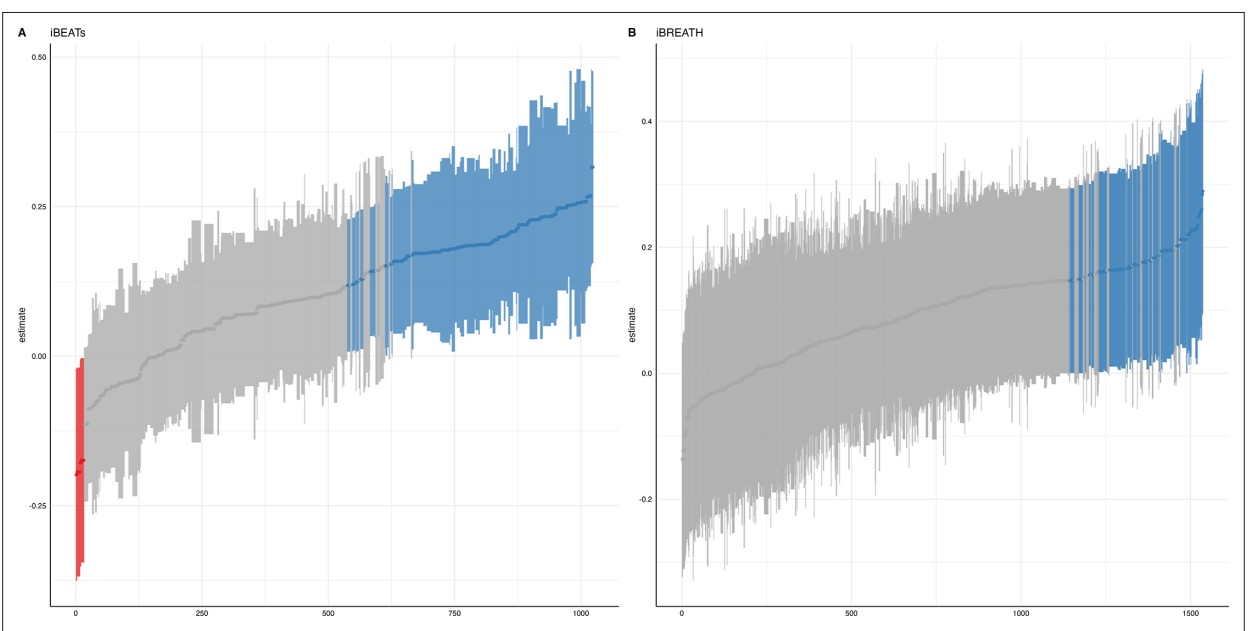

**Figure 5.** Specification curve analysis for the (**A**) iBEATs and (**B**) iBREATH task. Specification curve analysis plotting standardized beta regression coefficients (y-axis) and number of analysis (x-axis) for (**A**) iBEATs and (**B**) iBREATH. Number of analysis (x-axis) are ordered increasing from lowest to highest standardized beta regression coefficient. Blue color indicates a significant effect (p<0.05) for a mean synchronous preference, red color indicates a significant effect (p<0.05) for a mean asynchronous preference, and gray indicates a non-significant outcome.

analysis paths would have come to a different conclusion might indicate that the influence of analytical choices is not completely negatable.

## Discussion

In the present study we investigated cardiac and respiratory interoceptive sensitivity in 3-, 9-, and 18-mo-old infants utilizing a preregistered approach, validated by a specification curve analysis, and MEGA analytic approaches. We used two preferential looking paradigms that had a fixed trial structure. Regarding cardiac interoceptive sensitivity, we found evidence for a preference for stimuli presented synchronously in all three age groups. Regarding respiratory interoceptive sensitivity we find a more nuanced picture with infants showing a significant preference for stimuli presented synchronously at 9 mo of age, but not at 3- and 18 mos. We did not find strong evidence for a relationship between cardiac and respiratory interoceptive sensitivity in infants in the first year of life. However, we find some evidence for a positive relationship at 18 mo. Furthermore, in an exploratory analysis, we find indications that respiratory perception increases between 9- and 18 mo. However, due to the small sample size at 18 mo the results regarding changes and stability of interoceptive sensitivity in the second year of life must be considered speculative and need to be validated in further research.

In recent years, new theoretical frameworks have deepened our understanding of interoceptive processing (*Murphy et al., 2020*; *Suksasilp and Garfinkel, 2022*). For instance, in their 2x2 model, *Murphy et al., 2020* distinguished between two main factors of interoception – interoceptive accuracy (i.e. how exact one perceives internal bodily signals) and interoceptive attention (i.e. how often one thinks of internal bodily signals in everyday life). When applying the Murphy model to the iBEATs and the iBREATH, both aspects might be needed to show a preference. First, it is necessary to access one's own internal bodily signals to notice a difference between synchronous and asynchronous signals (i.e. interoceptive accuracy). Second, one also needs to pay attention to one's own bodily signals and compare them to what is happening on the screen (i.e. interoceptive attention). Thus, it is possible that the present task does not distinguish between both dimensions. Instead, the present task might measure a propensity to engage with own interoceptive signals (*Murphy, 2024*). In fact, when considering the potential impact of interoceptive sensitivity in real-world settings it is unlikely that 'pure' interoceptive accuracy or attention can be differentiated, but that the interplay of both shapes the outcomes.

We do not find evidence for a strong relationship between cardiac and respiratory interoceptive sensitivity in the first year of life. This finding is in line with empirical results not finding a relationship between cardiac and respiratory interoception in adults (*Garfinkel et al., 2016*) and children (*Nicholson et al., 2019*). Furthermore, these findings might be explained by accounts proposing different brain networks for processing of cardiac and respiratory information (*Suksasilp and Garfinkel, 2022*). Still, we find a relationship between cardiac and respiratory signals in the oldest sample tested here, the 18-mo-olds, which is closest to adults. Although this effect needs to be interpreted with caution due to the small sample size, this might indicate that using conceptually similar experimental paradigms might be a promising avenue to investigate relationships between different interoceptive modalities in adults.

To investigate individual differences, we used absolute proportional scores, following previous approaches (see *Figure 4*, *Maister et al., 2017*; *Weijs et al., 2023*). As a preference in any direction in the iBEATs or the iBREATH task can, in principle, be considered as evidence for the participant's ability to distinguish their own bodily signals from noise. However, it remains an open question if individual looking preferences for synchronous or asynchronous stimuli have a functional importance. In other studies, investigating infants' processing of information about body ownership, preferential looking paradigms similar to the iBEATs and iBREATH have been used. For instance, newborns prefer to look at synchronous visuo-tactile cues compared to asynchronous ones (*Filippetti et al., 2013*), similar to 7- and 10-mo-olds (*Zmyj et al., 2011*). In other cases, older infants showed a looking preference for sensorimotor-incongruencies (*Rochat, 1998*). Furthermore, at 5 mo of age infants recognize delays in visualization of their own leg-movements and prefer to look at stimuli that are asynchronous to their own movements (*Bahrick and Watson, 1985*). Thus, previous results are inconclusive as to whether infants generally prefer to look at stimuli that are synchronous or asynchronous to their own bodily movements and experiences. Yet, there is convincing evidence that they detect such (in-)

congruencies. For instance, 14-mo-old infants are more likely to help a person who had previously bounced in synchrony with them, compared to an asynchronously bouncing person (*Cirelli et al., 2014*).

Longer looking times for synchronous stimuli might indicate a familiarity preference (or more generally a preference for a stimulus that is easier to process). In this context, familiarity might refer to the infant's perception of congruence between internal signals and external stimuli which might drive the infant's attention. Specifically, the synchronous condition should be easier to process due to the intersensory redundancy and predictability between interoceptive and exteroceptive signals. Longer looking times for asynchronous stimuli might indicate a novelty preference, that is, a preference for a stimulus that offers a learning opportunity (*Hunter and Ames, 1988*). According to the framework presented by *Hunter and Ames, 1988*, the preference for novelty or familiarity depends on three factors that interact with each other: familiarization, age, and task difficulty. In short, the model proposes that less familiarization, a lower age, as well as an increase in task difficulty, facilitates a familiarity preference. However, it is important to consider that other cognitive and attentional mechanisms could also influence these responses.

Thus, when applying the framework to the present results it might be that certain details contributed to a familiarity preference displayed by most infants (as indicated by theaverage synchronous preference). For instance, the data presented here at 9- and 18 mo was collected as part of a larger study with several other paradigms, whereas the study of *Maister et al., 2017* used the iBEATs as the first task of the session. Thus, this increased complexity for the infant in our setting might have impacted the task difficulty and potentially reduced the familiarization. Furthermore, at 3 mo of age the experimental setting might be more challenging thus leading to an increased complexity. However, the interpretation of looking time preferences in infancy research in general and the *Hunter and Ames, 1988* framework specifically, remains a topic of debate and further research (*Bergmann et al., 2019*; *ManyBabies, 2023*).

Nevertheless, the switch in mean preference reported here regarding cardiac interoceptive sensitivity, compared to *Maister et al., 2017*, might also indicate a development around 5 mo of age. Infants at 5 mo of age might be more drawn to asynchrony between their cardiac signals and visual stimuli, while 3- and 9-mo-old infants on average prefer synchrony. Such a developmental trajectory might also explain the null findings reported by *Weijs et al., 2023*, as infants tested in their study were

**Table 6.** Number of significant results for specifications for iBEATs and iBREATH.

| Specification | iBEATs | iBREATH |
|---|---|---|
| Outlier rejection | 128, only async<br>330, both | 111 only async<br>158 both |
| SD outlier rejection | 107, no rejection<br>107, 2SD<br>120, 2.5SD<br>124, 3SD | 59 no rejection<br>70 2SD<br>104 2.5SD<br>36 3SD |
| Artifact trial rejection | 224, 85% criterion<br>162, small artifacts included<br>72, strict rejection | 207, large artifacts included<br>62, small artifacts included<br>0, strict rejection |
| Data transformation | 240, log transformed<br>218, not transformed | 162 log transformed<br>102 not transformed |
| Trial removal | 308, 0LTs included<br>150, 0LTs excluded | 183, 0LTs included<br>86, 0LTs excluded |
| Min. number of trials per id to be included | 115, min. 2 trials<br>115, min. 4 trials<br>113, min. 8 trials<br>115, no criterion | 86, min. 2 trials<br>74, min. 4 trials<br>24, min. 8 trials<br>85, no criterion |
| Statistical analysis | 220, linear mixed model<br>238, paired t-test | 89, linear mixed model<br>180, paired t-test |

Number of significant specifications for iBEATs and iBREATH separate for each category. Overall, there were 1024 specifications for iBEATs and 1536 for iBREATH. Detailed information on the specifications can be found in Appendix 4. Category 3 (artifact trial rejection) for the iBREATH was simplified to make comparison to the iBEATs more intuitive.

in between 5 and 7 mo of age. It is also possible that there are developmental windows in which the perception of bodily signals plays an important role. For instance, age groups in the present study were chosen to be in a similar range as the emergence of relevant theoretical constructs, such as mirror self-recognition. However, to disentangle such effects adequately powered longitudinal studies are needed.

To validate the impact of analytical choices on mean group preferences we used a specification curve analysis. In the following, we will discuss impactful decisions and make recommendations for future approaches (see *Table 6*). Regarding analytical choices that had an impact on the results, we found that applying the same physiological data rejection criteria to synchronous and asynchronous trials led to more significant results (*Table 6*, first entry). The logic behind not removing asynchronous trials with physiological artifacts in the tasks described here is that in these trials the signal is not generated by real-time feedback of the physiological signal. Thus, it is not directly relevant for stimulus presentation. However, our results indicate that applying differing criteria for both trial categories might obscure effects.

Moreover, we found that for both tasks, in terms of physiological artifact rejection, including more data points led to more significant results (*Table 6*, third entry). This might be explained by the inclusion of more data, and thus, greater statistical power. For instance, in the iBEATs task, strict artifact rejection means that a trial is removed once a single R-peak is not (or falsely) detected. However, in such trials, it might still be possible to recognize that the stimulus presentation is synchronous or asynchronous to one's heartbeat and it thus still holds information relevant for the task. For future studies, we would recommend a more fine-tuned approach for removing trials based on physiological artifacts. Furthermore, we would advise employing the same criteria to all conditions used.

Regarding specifications that did not have a strong impact, we found that outlier criteria using standard deviations had a negliable impact on the results (*Table 6*, second entry). Such criteria are usually applied to remove extreme values in the data. In the paradigms described here, looking times were bound by trial length (e.g. in the iBEATs max. 20 s). Thus, rejecting trials based on standard deviation might not be useful in analyses of preferential looking paradigms that use maximum trial length. One reason might be that extremely large outliers in looking times are impeded already by the experimental design. We also did not find that an inclusion criteria regarding a minimal number of valid trials an infant had to contribute to be included in the analysis changed the number of significant results much. Such criteria are typically used to increase the reliability of results, as individual trial outliers weight stronger when an infant only completes few trials. For instance, in our preregistered analysis, infants had to complete a minimum of eight trials for the iBEATs or four trials for the iBREATH to be included in the analysis (*Table 6*, sixth entry). However, there were few infants who completed less than these minimum number of trials. For future approaches, we would advise against using exclusion criteria based on standard deviations or number of trials. Moreover, the statistical test used (paired t-test vs linear mixed model, *Table 6*, seventh entry) had a rather small impact on the results. However, given the large number of analyses conducted, this might be related to not being able to precisely formulate the model to fit the complexity of the data for each specification.

Overall, the recommendations outlined above can be discussed within the scope of a fundamental challenge in experimental research – how to balance noise in a given dataset with losing statistical power by exclusion of participants and trials. This is especially relevant in infancy research that oftentimes deals with high drop-out rates and noisy datasets. For the present dataset, we find that leaning on the side of including more data points (e.g. regarding rejection of physiological artifacts, or exclusion criteria) might be more beneficial as long as the same criteria are applied to all data. Thus, exclusion of data points should be driven by trying to minimize the impact of erroneous or random datapoints, while still keeping those that have interesting characteristics (*Leys et al., 2019*). We want to stress that outlier criteria should ideally be formulated within a preregistration (*Bakker and Wicherts, 2014*).

Overall, we found more significant results for a group mean preference for the iBEATs (44.73%) compared to the iBREATH (17.51%) at 9 mo of age. Given that our exploratory analysis indicated an increase of iBREATH difference scores from 9 to 18 mo, respiratory interoceptive sensitivity might develop in this age range. However, it is also possible that the coupling of physiological signals with visual stimuli in infancy might produce stronger mean preferences for cardiac-, compared to

respiratory signals. In sum, the results of the specification curve analysis validated our preregistered analysis, as almost all analysis paths resulted in a numerical mean preference for synchronous stimuli.

## Ideas and speculation – development of respiratory interoceptive sensitivity

While we found consistent evidence for cardiac interoceptive sensitivity, whereby infants on average prefer stimuli presented synchronously with their heartbeat, the evidence regarding respiratory interoceptive sensitivity was more nuanced. In particular, the 18-mo-olds sample for the iBREATH displayed three interesting characteristics: it was the only sample showing a (numerical) preference for the asynchronous condition, absolute proportional scores increased compared to 3, and 9 mo, and there was a positive relationship with cardiac interoceptive sensitivity scores at 18 mo (but not at 3 or 9 mo). To interpret these results, one might speculate that a maturation of respiratory interoceptive sensitivity towards 18 mo of age takes place. A hypothesis to be tested in future research is that developmental improvement in respiratory perception might be related to increases in other domains that show links to interoception. For instance, self-perception matures towards the second birthday and has been conceptually related to interoception (*Fotopoulou and Tsakiris, 2017*; *Musculus et al., 2021*). Furthermore, gross motor development may be considered in future research, which drastically matures in the first two years of life (*WHO Multicentre Growth Reference Study Group, 2006*) and has been shown to be related to respiratory function in children with cerebral palsy (*Kwon and Lee, 2014*). However, the result and interpretation warrant further follow-up given the small sample size of the 18-moolds and exploratory nature of the respective analysis.

## Limitations

The data presented in this paper holds several limitations. First, due to an error in our experimental scripts, we unintentionally used a fixed-order design, instead of a semi-randomized/randomized design, in which almost all infants saw the same fixed order of stimuli (see Appendix 1 for additional analyses). Following, condition (always starting with a synchronous trial), and image assigned to condition was fixed. Furthermore, for a given trial the location of a stimulus (left/right) was fixed, although across all trials all condition/stimulus pairings were appearing on the left and right side of the screen equally often. Such a fixed-order design holds several important limitations as visual preferences might be influenced by the experimental design, i.e., the first trial always being synchronous might have influenced a mean group preference. Furthermore, we cannot rule out that mean group preferences were influenced by the stimuli used (as in most cases the same stimuli were used for synchronous/asynchronous trials) or by the location of the image in a given trial (left/right). Still, as the stimuli used were selected to be similar to each other, we would not expect that they would evoke a priori preferences. To further illustrate the impact of the fixed-order design we have conducted several additional analyses, which can be found in Appendix 1, which do not indicate that there was an impact of the fixed-order design. Specifically, we find no evidence for systematic differences between infants tested with the fixed design and infants tested with a randomized design.

However, we cannot fully rule out that the experimental design impacted infants' visual preferences. For instance, infant processing speed (*Hendry et al., 2016*), selective attention (*Reynolds and Romano, 2016*), or preference for visual salience (*Stallworthy et al., 2020*) differs across ages. Following, we cannot fully rule out that such factors might have in part driven infants' looking behavior. Still, the present results need to be considered in the context of the already existing literature on infant interoceptive sensitivity (*Maister et al., 2017*; *Imafuku et al., 2023*; *Weijs et al., 2023*). In addition, the results reported here provide rather stable evidence across ages. Therefore, despite potential confounding factors, it is likely that the interoceptive manipulation had an impact on infants' looking behavior (see also Appendix 1).

Despite these limitations fixed-order designs also hold advantages, as they are more suitable to investigate individual differences (*Dang et al., 2020*; *Hedge et al., 2018*). When each participant is exposed to the same procedure, individual differences are less likely to be attributed to effects of randomization but are more likely to reflect real differences between participants. Also, when considering the impact of the randomization, one must consider our results in relation to earlier studies (*Maister et al., 2017*; *Weijs et al., 2023*; *Imafuku et al., 2023*), some of which used the exact same stimuli as we did (*Maister et al., 2017*), with semi randomized designs. Results of these

studies indicate no looking times differences depending on the stimulus assigned to each condition or systematic preferences for one of the stimuli.

Furthermore, drop-out numbers must be discussed. For the 9-mo-old sample, ninety mother-infant dyads were invited to take part in the present study, but only 74/75 provided data for iBEATs and iBREATH, respectively. Furthermore, only 52 (iBEATs) and 56 (iBREATH) could be included in the confirmatory analysis based on the predefined exclusion criteria, and only 34 contributed usable data for both paradigms. This might also be attributable to the paradigms being embedded in a data collection for a larger project. Similarly, for the 3-mo-old sample 80 infants were invited to the lab, however, only 53 (iBEATs) and 40 (iBREATH) could be included in the analysis. Also, the 9- and 18-mo-old samples were collected during the Covid-19 pandemic, which led to high dropout rates for the longitudinal follow-up at 18 mo, as lockdowns and Covid-19 cases made data collection challenging. Thus, we might not have had sufficient statistical power to detect possible effects using our longitudinal sample. The reduced sample size might have impacted the statistical power to detect mean preferences for some age groups. Still, it must be noted that even the smaller sample sizes included were of similar size as used in previous studies on infant interoceptive sensitivity (*Imafuku et al., 2023*; *Maister et al., 2017*; *Weijs et al., 2023*).

To overcome some of these limitations, we have computed exploratory analysis using all data available, not just those infants that contributed data at both timepoints. However, such an approach can only provide correlational evidence. Regarding the specification curve analysis, it is possible that there are specifications that might be relevant, which were not considered here. Furthermore, in the specification curve analysis, we did not inspect assumptions underlying the statistical tests in-depth.

## Conclusion

To sum up, we present evidence that infants are sensitive to their own cardiac signals in the first two years of life using an adapted version of the paradigm introduced by *Maister et al., 2017*. Moreover, we present the first evidence that infants are sensitive to their respiratory signals using the iBREATH paradigm. By using a preregistered approach, a comparably large sample size and age range spanning the first two years of life, and by extending the interoceptive modality assessed to respiration, we provided important empirical evidence for theoretical accounts highlighting the relevance of interoceptive sensitivity in infancy. Regarding longitudinal development, we found no evidence for a change of interoceptive sensitivity in our confirmatory longitudinal analysis. However, exploratory analysis using a between groups approach revealed evidence for an age-related increase in respiratory, but not cardiac, interoceptive sensitivity scores towards 18-mo-of age. We did not find that cardiac and respiratory interoceptive sensitivity are strongly related, mirroring results in adults and children. However, we find exploratory evidence for a relationship at 18 mo.

We used a specification curve analysis to validate our results and showed that a specification curve analysis is a suitable tool to investigate the impact of analysis choices in infancy research. Finally, we provided guidelines for the analysis of the two paradigms presented here, as well as for preferential looking time paradigms in general. Overall, our results demonstrate that infants' interoceptive sensitivity, measured through coupling a visual presentation to a physiological signal, is a replicable phenomenon, that can be generalized to different age groups, as well as to different interoceptive modalities. However, it must be noted that the interpretability of the results is impacted by the fixed-experimental design, which is due to an error in the experimental scripts. By providing empirical results that go beyond previously published studies on infant interoception, our results give an important empirical basis for theoretical approaches targeting interoception during development, as well as related constructs such as self-perception, in early infancy.

## Materials and methods
### Sample

The data reported here was collected as part of a larger project involving a range of other measures. To stay coherent, we refer to each age group throughout the manuscript with regard to the lower end of the age range in which we included infants (e.g. we tested infants between 9 and 10 mo, but refer to them as the 9-mo-old group). For the 9-mo-old sample in total, 90 infant-mother dyads were tested in the laboratory. Initially, we intended to invite mother-infant dyads when the infant was 9–10 mo

of age. However, as this study was conducted during the Covid-19 pandemic, we extended the age range to 10 mo and 15 d to be able to include a sufficient number of infants ($M_{age}$ = 301.63 d, $SD_{age}$ = 10.57). We followed up the same sample again when the infants were 18–20 mo of age (N=54, $M_{age}$ = 576.65 d, $SD_{age}$ = 14.49). Data collection took place during the Covid-19 pandemic, from September 2020 to September 2021. The total sample size was based on a power analysis for an unrelated analysis. However, building up on the results reported by *Maister et al., 2017*; paired t-test; $t$=3.267, n=29, Cohen's d=0.4, the study would have been adequately powered to detect an effect approx. 30% (Cohen's d=0.3) smaller than reported by *Maister et al., 2017*. The 3-mo-old sample was tested after completion of the 9- and 18-month-old samples. Initially, we had planned to start data collection with the 3-mo-old sample. However, due to the Covid-19 pandemic, this was not possible. We invited 80 infant-mother dyads to the lab when the infant was 3–4 mo old ($M_{age}$ = 113.53 d, $SD_{age}$ = 7.82).

Participants were recruited from an existing database of volunteer families and parents. We strived to include an equal number of boys and girls. All infants were born full term with normal birth weight and had no known developmental delays or neurological impairments. Experiments were approved by the ethics committee of the University of Vienna (reference no. 00504).

## Experimental procedures

Upon arrival in the laboratory, primary caregivers were asked to fill out an informed consent form. After a warm-up period, the infants performed several tasks in randomized order. In the current manuscript, we only report results from the iBEATs and the iBREATH tasks, as results from the other tasks will be presented in separate reports. The order of the tasks was counterbalanced across participants. As both the iBEATs and the iBREATH followed a similar structure and required similar equipment, the tasks were performed back-to-back in an alternating order. Between the iBEATs and the iBREATH, we additionally acquired 3 min of resting state data to analyze cardio-respiratory coupling while infants watched a neutral video. The procedure was the same for infants from all age groups. Notably, the infants participating at 3 mo only did the iBEATs and iBREATH in alternating order.

## iBEATs

To measure cardiac interoceptive sensitivity, we used the iBEATs paradigm (*Maister et al., 2017*). Three electrodes were attached to the infant's chest in a three-lead setup. We used an ADInstruments Powerlab and BioAmp equipment to monitor and to record cardiac activity (https://www.adinstruments.com/). To identify R-peaks, we used the built-in hardware-based function, namely 'fast response output', which sends a pulse to a presentation-computer via a custom-made Arduino set-up, once a predefined threshold is reached. The threshold was set individually for each infant.

Upon the placement of ECG electrodes and the adjustment of the fast response output, infants were placed in an infant chair roughly 60 cm away from an eye tracker sampling at 500 Hz (Eyelink 1000 plus). The caregiver was asked to sit right behind the infant. In case the infant got fussy or did not tolerate the infant chair, we offered the option to place the infant on the caregiver's lap. Following a 3-point calibration, infants were presented with trials in which visual stimuli (i.e. either a yellow cloud or a pink star) moved rhythmically up or down on the screen, either synchronous or asynchronous to the infant's heartbeat. Movements of the stimuli were accompanied with a jumping sound to attract infants' attention. In synchronous trials, the movement of the stimulus on the screen was coupled to each infant's R-peak. For asynchronous trials, first, mean inter-beat-interval of the preceding synchronous trial were computed for each infant. Movement of the stimuli then followed a predetermined rhythm that was either 10% faster or slower than the average inter-beat-interval of the last synchronous trial for that infant.

There was a maximum of 80 trials in the task. The first trial was always synchronous. Before each trial, an attention getter was displayed. Once the infant looked at the screen, a trial started lasting for a minimum of 5 s and a maximum of 20 s. After the initial 5 s, the duration of the trials was infant controlled. The ongoing trial automatically terminated, and the next trial started, if the infant looked away from the screen longer than two consecutive seconds or the maximum trial duration of 20 s was reached. The task was terminated, if the infant looked away from the screen longer than four consecutive trials (i.e. a total duration of 20 consecutive seconds) or the infant became fussy or tired.

Initially, we had intended to counterbalance the stimuli across experimental conditions and infants. However, when conducting additional analysis during the review process we noticed an error in our

randomization scripts (for a detailed description and additional analyses regarding its impact see Appendix 1) which led to the iBEATs following a fixed-order experimental design for most participants across all age groups. Thus, for all infants apart from a small subsample of 9-mo-olds randomization was fixed, so that the same stimulus was synchronous or asynchronous, with a fixed order of locations (synchronous and asynchronous stimuli appeared both left and right stimuli appeared left and right) and a fixed trial order. However, the randomization was not completely alternating, so that in some cases two synchronous or asynchronous trials could follow each other. The stimulus presentation was performed using a custom-made script in MATLAB (Matlab 2018b).

## iBREATH

To measure respiratory interoceptive sensitivity, we developed and used the iBREATH paradigm, which followed a similar logic to the iBEATs task. A respiratory belt connected to an ADInstruments Powerlab was attached to the infant's torso (https://www.adinstruments.com/). Once a stable signal was obtained, infants were seated in an infant chair roughly 60 cm away from an eye tracker sampling at 500 Hz (Eyelink 1000 plus). The caregiver was asked to sit right behind the infant. The signal of the respiration belt was sent to a presentation computer using a custom-made Arduino set-up.

Similar to the iBEATs procedure, during a 3-point calibration, infants observed moving circles accompanied by a sound. Following calibration, infants were presented with an infant-friendly neutral stimulus (i.e. a red strawberry or a green apple), which increased and decreased in size, either synchronous or asynchronous to that infant's respiratory rhythm. Stimuli presentation was accompanied by an infant-friendly sound. The volume of the sound was adjusted in relation to the size of the stimuli, thus, increasing and decreasing as the image got bigger and smaller, respectively. In synchronous trials, the stimuli on the screen expanded and shrank in synchrony with each infant's respiration rhythm. In asynchronous trials, movement of the stimulus was either 10% faster or slower than the average breathing frequency of the last trial for that individual infant.

To generate asynchronous trials, two components of the immediately preceding synchronous trial were used to compute a sinusoidal signal that was either 10% faster or slower than the signal in the previous trial. First, the average breathing frequency of the last trial was extracted, which was either speeded up or slowed down based on the asynchronous trial type. Then, the average respiratory amplitude in the last synchronous trial was extracted, which was used to set the amplitude of the asynchronous trial. By combining frequency and amplitude, the sinusoidal signal was created.

The iBREATH paradigm consisted of a maximum of 80 trials. The first trial always was synchronous. Before each trial, an attention getter was displayed. A trial was displayed for a minimum of 5 s and a maximum of 30 s. Following the initial 5 s, the duration of the trials was infant controlled. An ongoing trial was terminated automatically, and the next trial started, if the infant looked away from the screen longer than two consecutive seconds or when the maximum trial duration of 30 s was reached. The task was terminated, if the infant looked away from the screen longer than four consecutive trials (i.e. a total duration of 20 consecutive seconds) or when the infant became fussy or tired.

Initially, we had intended to counterbalance the stimuli visual across experimental conditions and infants. However, when conducting additional analyses during the review process we noticed an error in our randomization scripts (for a detailed description and additional analyses regarding its impact see Appendix 1) which led to the iBREATH following a fixed-order experimental design for most participants across all age groups. Thus, for all infants apart from a small subsample of 9-mo-olds randomization was fixed, so that the same stimulus was synchronous or asynchronous, with a fixed order of locations (stimuli appeared left and right) and a fixed trial order.

## Confirmatory analysis

This study was preregistered on aspredicted.org. The preregistration for the 9-mo-old sample can be accessed here: https://aspredicted.org/QP9_6FP. The preregistration for the longitudinal analysis can be assessed here: https://aspredicted.org/GMB_XCW. The preregistration for the 3-mo-old sample can be accessed here: https://aspredicted.org/44L_QKH. Data, analysis, and experimental scripts are available here: https://doi.org/10.17605/OSF.IO/JY5FE.

## Pre-processing

In a first step, we visually inspected each trial of the iBEATs and the iBREATH tasks to exclude trials in which stimulus presentation was impacted by technical problems or physiological artifacts. We excluded trials for technical problems if transmission of the physiological signal was interrupted during a trial (e.g. an electrode was removed, a cable got unplugged, etc.) or stimulus presentation was interrupted (e.g. there was a problem in connecting to the stimulus presentation computer).

Next, we excluded trials with physiological artifacts. In the iBEATs, we excluded a trial if not all R-peaks were picked up by the fast-response-output. In the iBREATH, we excluded a trial if movement or other technical artifacts were visible in the respiratory signal during a trial. Furthermore, in the iBEATs, infants were included if they completed a minimum of eight trials. In the iBREATH, we adapted this criterion as respiration is a slower signal than the heartbeat and maximum trial durations were longer. As this might result in fewer total number of trials in the iBREATH task as compared to the iBEATs task, we adjusted the cut-off number for the iBREATH task and included data of infants who completed a minimum of four trials in the analysis. For the longitudinal analysis, we used a less strict criterion to increase our potential sample size as outlined in our preregistration. Thus, infants were included when they completed at least 4 trials in either task.

## Pre-processing of looking times-data

We defined an area of interests (AOIs) based on the maximum coordinates of the animated character on the screen. We took the maximum movement range of the animated character and computed looking times in each trial as the summed duration of all eye-tracking samples falling in that AOI. Because we aimed to replicate the study by *Maister et al., 2017*, we followed the same analysis approach as they did in the original paper. Accordingly, we excluded trials with looking times two standard deviations away from the condition's (i.e. synchronous or asynchronous trials) group mean. To compare cardiac and respiratory interoceptive sensitivity, we computed individual discrimination scores defined as the absolute proportion of looking time difference between synchronous and asynchronous conditions, again following the procedure by *Maister et al., 2017*. For both tasks, we excluded trials with looking times of 0, as it is not clear whether infants did not look at the screen in these trials, or whether there were technical issues in these trials.

## Statistical analysis

All statistical analysis reported here were computed in *R Development Core Team, 2022* using the packages 'pwr' (*Champely, 2020*), 'TOSTER' (*Lakens et al., 2018*), 'ggstatsplot' (*Patil, 2021*), 'Bayes-Factor' (*Morey and Rouder, 2022*), 'specr' (*Masur and Scharkow, 2019*), 'lme4' (*Bates et al., 2015*), 'afex' (*Singmann et al., 2022*), 'psych' (*Revelle, 2022*), 'broom.mixed' (*Bolker and Robinson, 2022*), 'bayestestR' (*Makowski et al., 2019*), 'DHARMa' (*Hartig, 2022*), 'glmmTMB' (*Brooks et al., 2017*), and 'faux' (*DeBruine, 2023*). To compute the Stouffer's z indices for the specification curve analysis we used the function provided in *Simonsohn et al., 2020*.

Out of the 90 mother-infant dyads invited to participate in the study, for the 9-mo-old sample, 74 infants contributed any data for the iBEATs task and 75 to the iBREATH task. For the iBEATs task, three additional infants were excluded due to technical errors. Furthermore, following our preregistered analysis, two infants were excluded for the iBEATs task due to not reaching the minimum of eight trials, nine due to noisy ECG data, and eight due to the +/-2 SD outlier rejection criterion, leaving a final sample of 52 infants. In comparison, for the iBREATH task, 10 infants were excluded due to technical errors, three due to not reaching at least four trials, three due to noisy respiratory belt data, and three due to the +/-2 SD outlier rejection criterion, leaving a final sample of 56 infants.

As outlined in our preregistration, we lowered the threshold for outlier rejection in the longitudinal analysis to increase the sample size. Thus, for all analysis infants who completed at least four trials per task were included. For the 9-mo-old data, this would have slightly changed the iBEATs analysis plan. However, this criterion did not lead to the inclusion of additional infants in the final sample. For the 18-mo-olds' iBEATs data, no infants were excluded due to not reaching at least four trials, four infants were excluded due to quality of the ECG signal, and two infants were excluded due to the +/-2 SD outlier rejection criterion, resulting in a final sample of 28. For the 18-mo-olds' iBREATH data, one infant was excluded due to not reaching at least four trials, three infants were excluded due to noisy physiological data, and three infants were excluded due to the +/-2 SD outlier rejection criterion,

**Table 7.** Descriptive information for number of trials completed and included.

| Paradigm, Age group | $M_{completed}$ | $SD_{completed}$ | $M_{included}$ | $SD_{included}$ |
|---|---|---|---|---|
| iBEATs, 3 mo | 13.97 | 7.08 | 9.82 | 7.44 |
| iBEATs, 9 mo | 18.16 | 6.35 | 9.52 | 6.63 |
| iBEATs, 18 mo | 15.62 | 6.31 | 10.90 | 7.54 |
| iBREATH, 3 mo | 13.00 | 5.84 | 9.16 | 6.52 |
| iBREATH, 9 mo | 13.25 | 4.85 | 10.10 | 5.21 |
| iBREATH, 18 mo | 12.52 | 7.63 | 6.88 | 5.56 |

leaving a final sample of 30 infants. Means and SDs for number of trials completed for infants included in the analysis can be found in *Table 7*.

Out of the 80 mother-infant dyads invited to participate in the 3-mo-old study, 77 infants contributed any data for the iBEATs task and 71 to the iBREATH task. Furthermore, following our preregistered analysis, one infant was excluded for the iBEATs task due to noisy ECG data, and 23 due to problems with the eye-tracking giving a sample of 53 infants. In comparison, for the iBREATH task, two infants were excluded due to not reaching at least four trials, 10 due to noisy respiratory belt data, and 19 due to problems with the eye-tracking resulting in a final included sample of 40 infants.

All statistics in our confirmatory analysis using null hypothesis testing were evaluated against a two-tailed significance level of $p < 0.05$. In case of non-significant results, if possible we aimed at following up the respective analysis with an equivalence or region of practical equivalence test (*Lakens et al., 2018*). To compare synchronous and asynchronous trials at 9 and 18 mo, both for the iBEATs and the iBREATH tasks, we computed two separate paired t-tests (*Maister et al., 2017*; see Appendix 5 for more information on asynchronous trials). At 3 mo we used a Bayesian paired t-test as the data collection was done after having collected the 9- and 18-mo-old samples. Our intention in the analysis of the 3-mo-old sample was to focus on strength of evidence in favor of/against an effect instead of a binary classification. We preregistered to correlate iBEATs and iBREATH scores at 9 mo. However, in the manuscript, we only report the details of the MEGA-analysis (see next paragraph). To investigate the longitudinal development of cardiac and respiratory interoceptive sensitivity, we computed a Bayesian paired t-test comparing absolute proportional scores between 9 and 18 mo.

## MEGA-analysis

We computed three MEGA-analyses pooling together data from all three age groups – to investigate a mean preference effect, the relation between the iBEATs and the iBREATH, as well as the development over age groups. First, to investigate whether there is a mean preference in the iBEATs and the iBREATH tasks, we computed mixed models using the R-package 'glmmTMB' utilizing a beta-error distribution and logit-link function. We used looking time as outcome, condition, age, and their interaction as fixed effect, and participant as a random effect. We transformed age into a factor with 3-levels (3, 9, 18 mo), whereby 3 mo was set as reference level. After fitting the model, we visually inspected assumptions using the check_model function of the R-package 'performance'. In addition, we checked for overdispersion using the 'DHARMa' package (iBEATs: dispersion = 1.07, $p=0.168$; iBREATH: dispersion = 1.12, $p=0.120$). Furthermore, we checked a reduced model lacking the interaction term for issues of collinearity (iBEATs, VIF = 1.00; iBREATH: VIF = 1.00). We then conducted full-null model comparisons by fitting a null-model that excluded the condition factor.

To investigate whether there is a relationship between the iBEATs and the iBREATH absolute proportional scores we computed a mixed model using the R-package 'glmmTMB' using a beta error distribution. We used the iBREATH scores as outcome variable, the iBEATs scores, age group and the interaction as factors, and participant as a random intercept. Age was included as a factor with 3 mos as reference level. After fitting the model, we visually inspected assumptions using the check_model function of the R-package 'performance.' We also did not find evidence for overdispersion (dispersion = 1.03, $p=0.800$). Furthermore, we checked a reduced model lacking the interaction term for issues of collinearity (VIF = 1.04).

Last, to investigate whether there is a difference between absolute proportional scores in the iBEATs and the iBREATH we computed two mixed models using the R-package 'glmmTMB' with a beta error distribution. We used the iBEATs or the iBREATH absolute proportional scores as outcome, age as factor, and participant as a random effect. Age was included as a factor with three levels. After fitting the model, we visually inspected assumptions using the check_model function of the R-package 'performance'. Furthermore, we checked for absence of overdispersion (iBEATs: dispersion = 1.07, p=0.560; iBREATH: dispersion = 1.09, p=0.600). We found that for the iBEATs, the full model did not significantly improve fit over the null model ($\chi^2$(3)=0.170, p<0.919), but for the iBREATH, the full model did provide a significantly better fit than the null model ($\chi^2$(3)=10.60, p=0.005).

## Acknowledgements

This research was funded in whole or in part by the Austrian Science Fund (FWF) [Project Number: P33486]. For open access purposes, the author has applied a CC BY public copyright license to any author-accepted manuscript version arising from this submission. Ezgi Kayhan was funded by the DFG (Project number: 402789467). Manos Tsakiris was supported by the European Research Council Consolidator Grant (ERC-2016-CoG-724537) under the FP7. We want to thank all infants and mothers who participated in this project. We also want to thank Monica Vanoncini and Liesbeth Forsthuber, as well as all research assistants, interns, and master students for their help in data collection and preparation of the experiment: Sandra Gaisbacher, Laura Neumann, Julia Otter, Lisa Triebenbacher, Jakob Weickmann, Felicia Wittmann, Gesine Jordan, Nina Maier, Rebecca Lutz, Celine Dorczok, Ann-Cathrine Gärtner, Maria Baumann, Nadine Pointner.

## Additional information

### Funding

| Funder | Grant reference number | Author |
| --- | --- | --- |
| Austrian Science Fund | 10.55776/p33486 | Markus R Tünte Stefanie Hoehl |
| Deutsche Forschungsgemeinschaft | 402789467 | Ezgi Kayhan |
| FP7 Ideas: European Research Council | ERC- 2016-CoG-724537 | Manos Tsakiris |

The funders had no role in study design, data collection and interpretation, or the decision to submit the work for publication.

### Author contributions

Markus R Tünte, Ezgi Kayhan, Conceptualization, Resources, Data curation, Software, Formal analysis, Supervision, Funding acquisition, Validation, Investigation, Visualization, Methodology, Writing – original draft, Project administration, Writing – review and editing; Stefanie Hoehl, Conceptualization, Resources, Supervision, Funding acquisition, Validation, Methodology, Writing – original draft, Project administration, Writing – review and editing; Moritz Wunderwald, Conceptualization, Software, Validation, Visualization, Methodology; Johannes Bullinger, Asena Boyadziheva, Validation, Investigation, Writing – original draft, Writing – review and editing; Lara Maister, Software, Validation, Methodology; Birgit Elsner, Conceptualization, Supervision, Funding acquisition, Project administration; Manos Tsakiris, Conceptualization, Supervision, Funding acquisition, Validation, Writing – original draft, Writing – review and editing

### Author ORCIDs

Markus R Tünte ⓘ https://orcid.org/0000-0002-4376-2908
Stefanie Hoehl ⓘ https://orcid.org/0000-0003-0472-0374
Birgit Elsner ⓘ https://orcid.org/0000-0003-3441-2436
Manos Tsakiris ⓘ http://orcid.org/0000-0001-7753-7576

## Ethics

Experiments were approved by the ethics committee of the University of Vienna (reference no. 00504).

Reviewer #1 (Public review): https://doi.org/10.7554/eLife.91579.4.sa1
Reviewer #2 (Public review): https://doi.org/10.7554/eLife.91579.4.sa2
Author response https://doi.org/10.7554/eLife.91579.4.sa3

## Additional files

### Supplementary files
MDAR checklist

### Data availability
Data, analysis and experimental scripts are available here: https://doi.org/10.17605/OSF.IO/JY5FE.

The following dataset was generated:

| Author(s) | Year | Dataset title | Dataset URL | Database and Identifier |
| --- | --- | --- | --- | --- |
| Tünte MR, Höhl S, Wunderwald M, Bullinger J, Boyadziheva A, Maister L, Elsner B, Tsakiris M, Kayhan E | 2024 | Respiratory and Cardiac Interoceptive Sensitivity in the First Two Years of Life | https://doi.org/10.17605/OSF.IO/JY5FE | Open Science Framework, 10.17605/OSF.IO/JY5FE |

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

# Appendix 1

## Randomization error in iBEATs and iBREATH scripts

When conducting additional analysis regarding trial order effects during the review process, we noticed an error in the randomization procedure of our experimental scripts, which were custom Matlab scripts (all scripts are publicly available in the corresponding osf project). The error stems from the use of Matlab functions designed to return random structures, such as the 'randperm' function, without the specification of a random seed. If a random seed is not specified before running these functions, they will always return predefined sequences that are coupled to the respective computer (i.e., the first 'random' sequence generated once starting Matlab will always be the same, and the second 'random' sequence generated will always be the same). Thus, when repeatedly running this command in Matlab without restarting Matlab, seemingly random sequences will be returned. However, the first time a script is ran when the program is started will always result in the same sequence.

Apart from a few testing sessions with 9-mo-olds we always restarted Matlab before every paradigm was run in order to make sure that the connection to the Arduino, which we used to synchronize physiological signals with the Eye tracking, was reset. The reason for this was that the Matlab would sometimes crash in case the connection to the Arduino was not established before every paradigm. This means, that in contrast to our intended experimental approach of randomizing stimuli location and image used, almost all infants watched the same combination of image used for synchronous and asynchronous condition, as well as order of location. Also, in contrast to our intended experimental approach for the 3-mo olds sample, in which we wanted to counterbalance synchronous and asynchronous conditions for the first trial, the first trial was always synchronous.

## Alternative randomizations

One potential issue with such a fixed-order design is that our experimental effects might have been impacted by the lack of randomization. As most infants were presented with the same image in synchronous or asynchronous conditions it is possible that mean group differences between conditions are influenced by the stimulus assigned to the respective condition (stimuli for iBEATs and iBREATH are displayed in *Figure 1*). However, for some infants in the 9-mo olds group we did not restart Matlab after every experimental paradigm. Thus, they were presented with a different randomization. To illustrate the impact of the randomization on our results we decided to repeat our main analysis reported in the manuscript for the 9-month-olds but split the samples apart according to their randomization (labeled as randomization 1 and 2 in the following). In addition, we did not reject trials using standard deviations as we wanted to retain as much data as possible. Results are displayed in *Table 1*. For the smaller samples we used a Wilcoxon ranked sign test instead of the paired t-test.

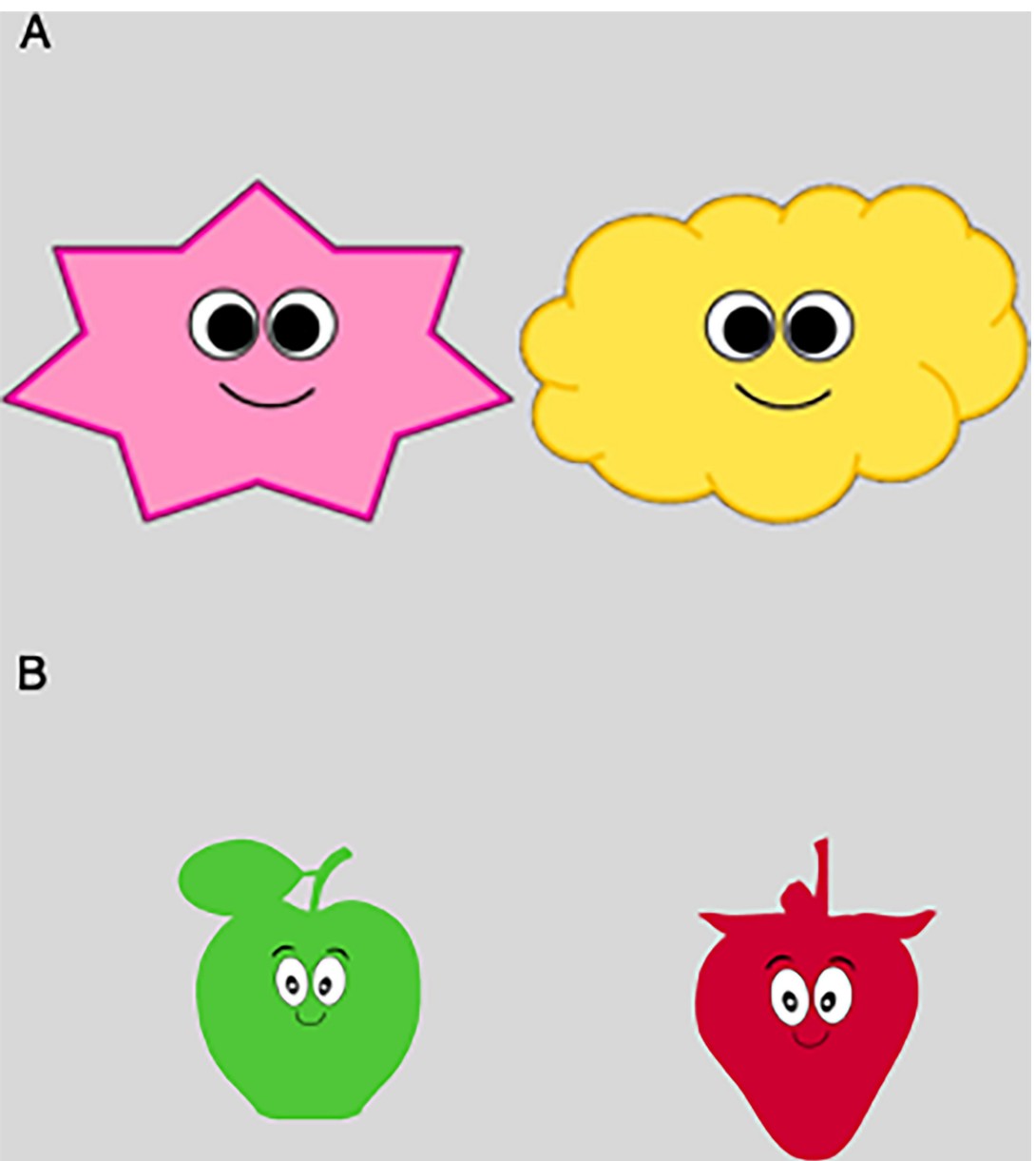

**Appendix 1—figure 1.** Stimuli used for (**A**) iBEATs and (**B**) iBREATH.

**Appendix 1—table 1.** Main analysis for the 9-month-olds for iBEATs and iBREATH split by randomization.

| Experiment | Randomization | N | Mean LT Synch (SD) | Mean LT Async (SD) | Comparison |
|---|---|---|---|---|---|
| iBEATs | 1 | 46 | 6792 (3473) | 5727 (2508) | t(45)=2.37, p=0.022 |
| iBEATs | 2 | 6 | 8775 (5810) | 3734 (1175) | V=21, p=0.031 |
| iBREATH | 1 | 53 | 6307 (3090) | 5563 (2282) | t(52)=2.37, p=0.021 |
| iBREATH | 2 | 3 | 6847 (1321) | 4085 (331) | V=6, p=0.25 |

LT = looking time, Synch = synchronous trials, Async = asynchronous trials. For the larger samples a paired t-test was used, and a Wilcoxon rank sign test for the smaller samples.

For the iBEATs we find that six infants, and for the iBREATH three infants, completed an alternative randomization (marked as randomization 2) and had enough data to be included in the analysis. Furthermore, when considering mean preferences, we find that the exclusion of the

infants with alternative randomization (so only considering randomization 1) does not change the results of our main analysis, as for both paradigms a significant mean preference for synchronous trials is still present. Next, when just considering the alternative randomization, we find that the numerical preference is the same for both iBEATs and iBREATH, with the iBEATs showing a significant mean preference for synchronous trials. Notably, these results need to be interpreted with caution due to the small sample sizes for randomization 2. Last, we aimed at investigating whether there is a significant difference between infants in randomization 1 or randomization 2 for iBEATs and iBREATH, respectively. To do so we computed a difference score for each infant by subtracting mean looking times in synchronous trials minus mean looking time in asynchronous trials. Then we used a Wilcoxon rank sign test to investigate whether the two groups differed from each other. We do not find evidence for a significant difference between randomization groups for iBEATs ($W=91$, $p=0.188$, *Figure 1A*), or iBREATH ($W=34$, $p=0.102$, *Figure 1B*). Following, these results indicate that the stimuli assigned to synchronous or asynchronous trials did not exhibit a impact on infant preference.

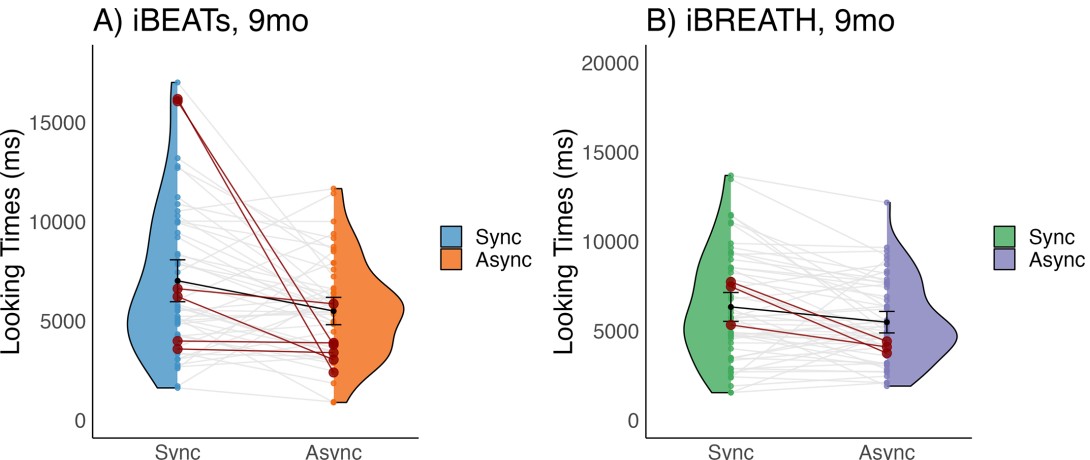

**Appendix 1—figure 2.** Mean differences in (**A**) iBEATs (blue refers to the synchronous and orange to the asynchronous condition) and (**B**) iBREATH (green refers to the synch and purple to the asynchronous condition) for the 9-mo-olds. Gray dots and lines refer to individual infants, black dots and error bars to the results of mean comparison and corresponding 95% confidence intervals. Infants with alternative randomization are highlighted in red.

## Preferences over time

Next, we aimed at investigating whether the infant's mean preference for synchronous or asynchronous trials changed over time in both tasks. For this analysis we used data from the confirmatory mean group preference analysis from all age groups. We computed two different analyses. First, in a cumulative approach (*Figures 3 and 4*, left column), we computed paired t-tests (and corresponding 95% confidence intervals) with a cumulative increasing number of trials included. This means that the first point displayed corresponds to the mean difference found in a paired t-test using only the first two trials, while the last point displayed corresponds to a paired t-test using all trials (our confirmatory analysis). This analysis gives us an indication of how the mean effect varies with the inclusion of an increasing number of trials.

In the second analysis we used a sliding window approach of three trials regarding trial inclusions (*Figures 3 and 4*, right column). This means that each point shows the result of a paired t-test (and corresponding 95% confidence intervals) using the three trials before and after the corresponding trial (e.g., for trial 10, trials 7–13 are included). This analysis gives us an indication of how the infant's preference changes throughout the paradigm.

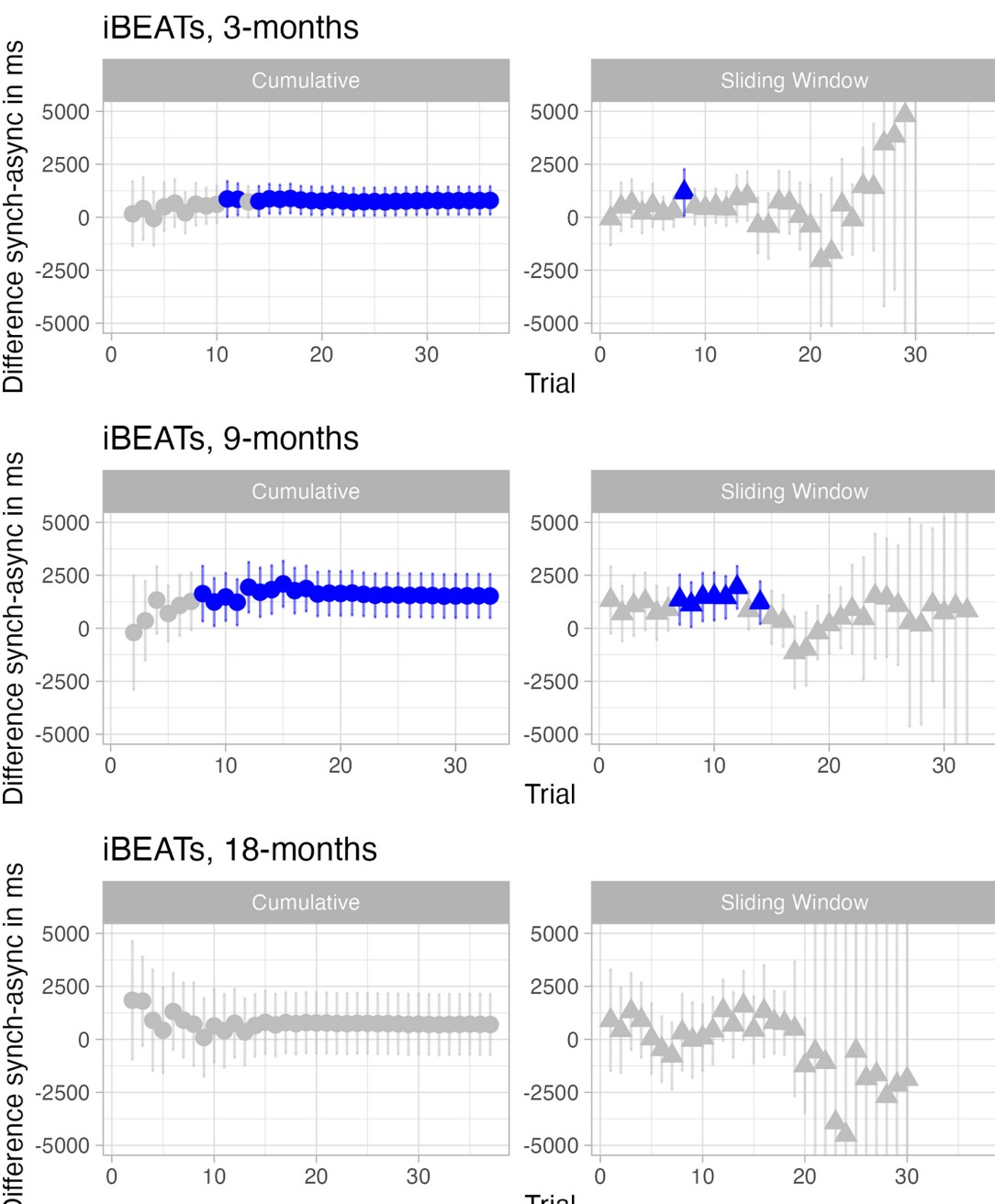

**Appendix 1—figure 3.** Infants' preference over time in the iBEATs paradigm for all age groups. Here results of paired t-tests for the iBEATs with different inclusion criteria for trials are displayed for all age groups. The left column shows a cumulative analysis in which all previous trials are included. The right column shows a 3-trial sliding window analysis in which the three preceding and three proceeding trials are included. Blue indicates a significant result of the paired t-test.

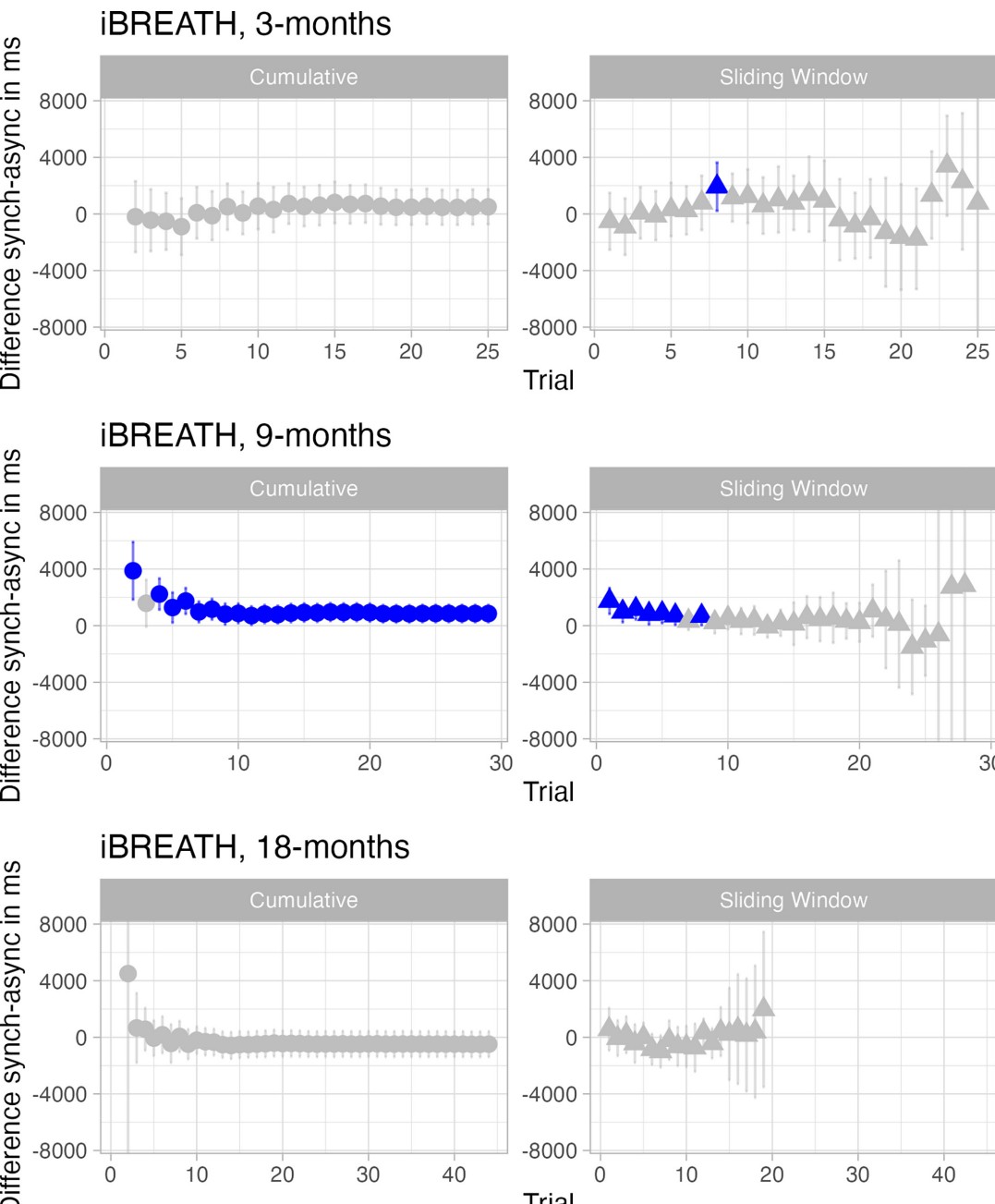

**Appendix 1—figure 4.** Infants' preference over time in the iBREATH paradigm for all age groups. Here results of paired t-tests for the iBREATH with different inclusion criteria for trials are displayed for all age groups. The left column shows a cumulative analysis in which all previous trials are included. The right column shows a 3-trial sliding window analysis in which the three preceding and three proceeding trials are included. Blue indicates a significant result of the paired t-test.

For the cumulative analysis (*Figures 3 and 4*, left columns) we find that for the iBEATs (*Figure 3*) in the 3- and 9-mo-olds mean preferences seem to stabilize around 7–8 trials, with the early trials not showing a clear preference. For the 18-mo-olds we find a stronger preference in the first trials, which then shifts towards no preference. For the cumulative analysis in the iBREATH (*Figure 4*) we find that in the 9- and 18-mo-olds the first two trials have an impact, while the mean preference stabilizes relatively fast and stays rather constant. In turn, for the 3-month-olds we do not see an impact of the first trial in the iBREATH.

For the sliding window analysis (*Figures 3 and 4*, right column) we find that for the iBEATs (*Figure 3*) in the 3- and 9-mo-olds the strongest preferences are around the trials 7–13. Interestingly,

this coincides with the trial numbers in which the cumulative analysis finds a stabilization of preferences. While for the 18-mo-olds we find that preferences are not stable and drift around. For the iBREATH (*Figure 4*) we find that rather strong preferences can be observed for the first trials, with later trials not showing strong effects. In general, for the sliding window analyses it must be noted that later trials show very unreliable effects with large confidence intervals. This is probably due to fewer number of infants included as mean numbers of trials completed ranged from 12.52 (iBREATH 18 mo) to 18.26 (iBEATs 9 mo).

Taken together, we find that the effect of the trials on mean group preferences varies both across tasks and age groups. If present, for the iBEATs (3-and 9-mo-olds) mean group effects seem to emerge in later trials (around trial 7), while for the iBREATH (9 and 18-mo-olds) the first trials were more impactful. At the same time, we do not see a coherent effect across all age groups, in the iBEATs the 18-month-olds, and in the iBREATH the 3-mo-olds, show more diffuse patterns.

If the randomization of the trials, and first trial being fixed, had had a big impact on infants' looking patters, similar patterns should have emerged across tasks and age groups. Given that we do not find such a coherent pattern, our results probably capture sufficient variation with regard to the interoceptive manipulation present in the tasks.

## First vs second half of presentation

Next, we provide descriptive statistics on looking times during the first and second half of stimulus presentation. For each infant we computed the categories first half and second half as the number of trials included in the MEGA analysis divided by two. This means that the halfs are computed individually per infant, e.g., for an infant that has eight included trials, first and second half would comprise four trials each. In case of an uneven number of trials we dropped the middle trial. Next, we computed mean looking times to the synchronous and asynchronous condition for the first and second half for each age group. Results are displayed in *Table 2* (iBEATs) and *Table 3* (iBREATH). In general, looking times are longer for the first half of trials compared to the second half of trials. Furthermore, looking times follow the trends found in our analysis of mean looking times. For groups with strong preferences in the analysis, also larger differences can be observed for conditions irrespective of the half of presentation (e.g. iBEATs, 9 mo, *Table 2*). For groups whose results showed more uncertainty, differences between conditions are less clear, while in general the first half has longer looking times than the second half (e.g. iBREATH, 18 mo, *Table 3*).

**Appendix 1—table 2.** Mean looking time for first and second half of trials in the iBEATs.

| Age | Half | Mean looking time (SD) synchronous | Mean looking time (SD) asynchronous |
|---|---|---|---|
| 3 mo | 1 | 7620 (4786) | 6686 (6193) |
| 3 mo | 2 | 6010 (5899) | 5306 (5443) |
| 9 mo | 1 | 8933 (4890) | 8197 (4954) |
| 9 mo | 2 | 5724 (4475) | 5089 (3992) |
| 18 mo | 1 | 8552 (5536) | 8575 (5772) |
| 18 mo | 2 | 5607 (4527) | 5402 (4786) |

**Appendix 1—table 3.** Mean looking time for first and second half of trials in the iBREATH.

| Age | Half | Mean looking time (SD) synchronous | Mean looking time (SD) asynchronous |
|---|---|---|---|
| 3 mos | 1 | 9188 (9176) | 8356 (8654) |
| 3 mo | 2 | 5903 (7876) | 6353 (7594) |
| 9 mo | 1 | 8272 (5597) | 7110 (5486) |
| 9 mo | 2 | 5774 (5425) | 5135 (4418) |
| 18 mo | 1 | 6659 (5578) | 6976 (6629) |
| 18 mo | 2 | 3550 (4124) | 4532 (3482) |

## Side preferences

Next, we considered whether the side of the presentation might have impacted infants' preferences. For each age group and condition we computed descriptive statistics for whether trials were displayed on the left or on the right side of the screen (*Table 4*). Location on the screen was randomized within the trial order, so all synchronous and asynchronous trials could appear on the left side and on the right side of the screen. But as outlined earlier, the order of the presentation was fixed, so whether a given trial was presented on the left or right was the same for most participants. From the descriptive statistics we see that there is some relation between side of presentation and condition, as well as mean looking times.

For instance, for the iBEATs in the 3 mo age group asynchronous trials were displayed more often on the left, and synchronous trials on the right. Furthermore, the mean preference in looking time seems to be driven by synchronous trials that are displayed on the right (*M*=7367 ms), while all other categories show similar looking times (Means from 5835–5978 ms). Thus, there is some indication that the side of presentation might have interacted with the infant's looking preferences. Still, overall, the side of presentation does not seem to have opposing effects on the looking time preferences in our data. The preferences for stimuli on the right side in the 3-mo-olds is only present for the synchronous condition (*M*=7367 ms) but not for the asynchronous condition (*M*=5978 ms). For the 9-mo-olds the descriptive statistics indicate that side of presentation had less of an impact. Overall, both in iBEATs and in iBREATH there are longer looking times to synchronous, than to asynchronous trials, irrespective of whether the stimulus was presented left or right (apart from the 18 mo iBREATH sample).

**Appendix 1—table 4.** Mean looking times (SD) in ms and number of trials for side of the screen (left/right), condition, and age groups.

| Age | Side | Condition | Number of trials iBEATs | Mean looking time (SD) in ms iBEATs | Number of trials iBREATH | Mean looking time (SD) in ms iBREATH |
|---|---|---|---|---|---|---|
| 3 | L | Synchronous | 116 | 5807 (5854) | 95 | 7140 (8689) |
| | L | Asynchronous | 185 | 5708 (5835) | 135 | 7138 (8230) |
| | R | Synchronous | 181 | 7367 (6130) | 147 | 7841 (8673) |
| | R | Asynchronous | 117 | 5978 (5775) | 81 | 7461 (7806) |
| 9 | L | Synchronous | 158 | 7137 (5066) | 87 | 6948 (6137) |
| | L | Asynchronous | 238 | 6962 (4963) | 197 | 6185 (5418) |
| | R | Synchronous | 242 | 7451 (4889) | 189 | 6567 (5235) |
| | R | Asynchronous | 173 | 5619 (4269) | 107 | 5642 (4416) |
| 18 | L | Synchronous | 69 | 7206 (5323) | 23 | 5099 (5157) |
| | L | Asynchronous | 108 | 6861 (5395) | 78 | 5433 (5567) |
| | R | Synchronous | 109 | 6867 (5286) | 82 | 5104 (5076) |
| | R | Asynchronous | 67 | 6904 (5753) | 41 | 5694 (4890) |

Values displayed here are based on the pre-processing pipeline used for the MEGA-analysis and are computed with unaggregated dataset. L=left, *R*=right.

## Conclusion

In sum, we conducted several analyses to investigate the impact of the experimental design on infant looking behavior. First, we considered those infants from the 9-mo-old subgroup with an alternative randomization. We do not find that the image used for synchronous or asynchronous trials had an impact on mean preferences in our analysis. Next, we considered the trial order and investigated whether we find patterns regarding the emergence of preferences. We find that infants' preferences vary over tasks and age groups. Last, we also considered whether the side of stimulus presentation impacted the infants' looking behavior. We find that the side of presentation might have mattered for the iBREATH. However, the main effect of condition across all samples stays the same. Overall, these analyses do not give strong indication that the randomization was a strong driver of the infants' looking behavior.

Neuroscience

Taken together, we do not find evidence that our results have been impacted by the use of a fixed experimental approach. Given that the images used in the experimental paradigms presented here were chosen with the intention of not inducing an a priori preference in the infants, this result is not necessarily surprising. Furthermore, as our initial intention was to use a semi-random paradigm, with the first trial always being synchronous, and never more than two trials of the same type following each other, it is also unlikely that the trial order used here is substantially different from one used in more randomized approaches. However, the fixed experimental design is a major limitation of the present study, and we cannot fully rule out that it influenced infants' looking behavior to some extent.

# Appendix 2

## Data simulation for different sample sizes

In our main analysis, we have found that some of the tests investigating mean difference between conditions were not statistically significant. Follow-up analysis using equivalence tests and region of practical equivalence approaches showed that sample size might have played a role, as statistical power could have been too low to detect an effect. Here, we aimed at further investigating the absence of a significant effect for a mean group preference due to reduced statistical power in smaller samples. To investigate this hypothesis, we decided to run simulations building up on our data. Our aim was to characterize how statistical power to detect an effect is impacted by different levels of sample sizes. This is relevant as sample sizes in infancy research tend to be low, and all non-significant results reported so far for iBEATs-like paradigms had a sample size of roughly 30 infants (*Weijs et al., 2023*, the 18-mo-old samples reported here). Furthermore, the results of such simulations might be very informative for researchers planning to use experimental paradigms like iBEATs or iBREATH in infant samples in the future.

In a first step, we used the R-package 'faux' to simulate data sets using the 9-mo-old data from the iBEATs as input. The package 'faux' simulates data that has the same properties as an existing data set. We simulated data sets ranging from 5 to 125 participants and generated 50 data sets for each number of participants, giving a total of 6000 datasets. We used the data sets processed according to our preregistration as input data set (n=52, Cohen's d=0.48).

Next, we ran analyses with the generated data which are visualized in *Appendix 2—figure 1*. We computed a t-test following our preregistered analysis strategy comparing mean looking times for synchronous and asynchronous conditions for different sample sizes. In 1 A mean differences for the paired t-test (y-axis) are plotted against the sample size for the simulated data obtained building up on our 9-mo-old sample. Red color refers to a significant result from the paired t-test, while blue color refers to a non-significant effect. In (B) the percent of significant results for the paired t-tests (y-axis) are plotted against the sample size (x-axis).

The results from the simulation give us an idea about the chance to find a significant result given an effect of d=0.48 in data sets with varying sample sizes. We observed that for the data building up on our 9-mo-old sample, upon approaching a sample size of 50–60, 80–90% of results are significant. Interestingly, the proportion of significant results for a sample size of 30 are a little bit above 50%, indicating that with a sample size of 30 infants the chance to find a significant mean difference is roughly that of a coin flip. Thus, the absence of a significant result in samples of 30 infants might not necessarily indicate the absence of a mean group preference in general, but the sample size might not have been sufficient to detect a significant result.

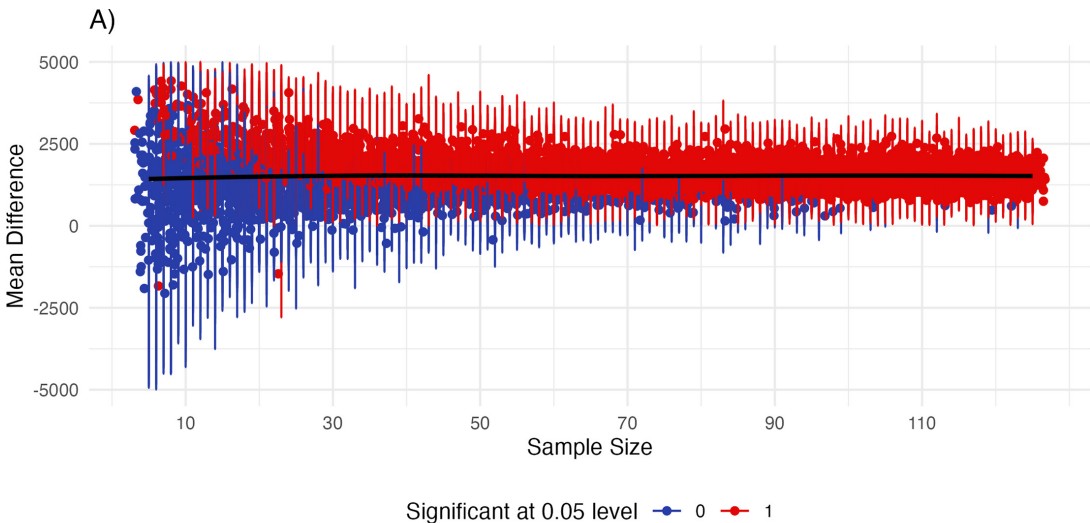

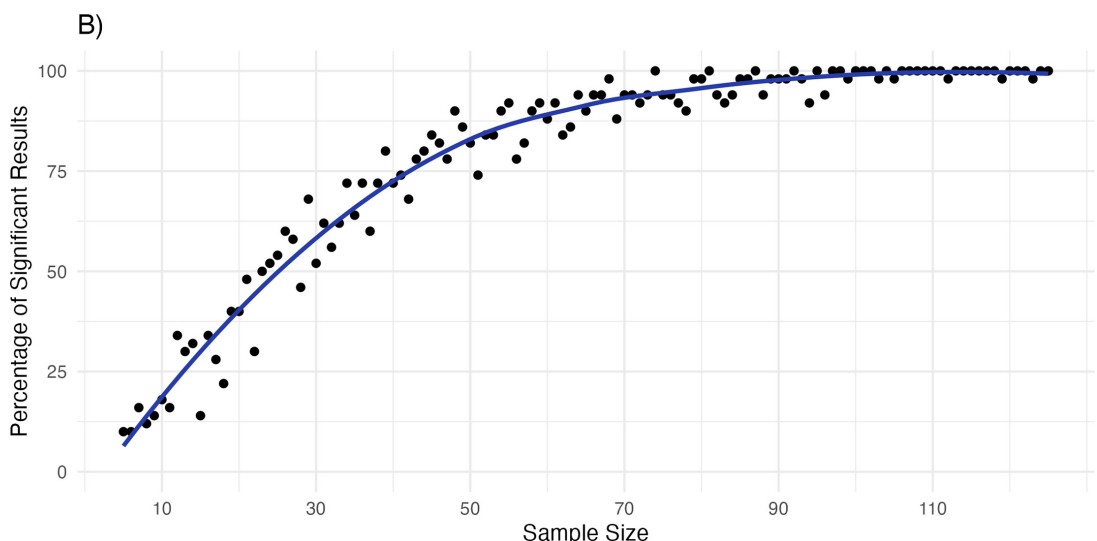

**Appendix 2—figure 1.** Simulating data frames for sample sizes from 15 to 125 building up on the iBEATs data from the 9-mo-olds. Results from the simulations. In (**A**) mean effects and 95% confidence intervals are plotted for the different sample sizes. Red color indicates a significant effect, while blue indicates a non-significant result. In (**B**) the percent of significant results are plotted with a fitted line.

## Appendix 3

## Detailed results for the MEGA-analyses

Here, we provide detailed results for the MEGA-analyses for the iBEATs and iBREATH. iBEATs

Table 1 reports the full-null model comparison for the iBEATs MEGA-analysis. Table 2 reports the coefficients for the iBEATs MEGA-analysis with 3 mo as reference group, while Table 3 reports the coefficients for the iBEATs MEGA-analysis with 9 mo as reference group and Table 4 with 18 mo as reference group.

**Appendix 3—table 1.** Full-null model comparison for the iBEATs model.

| Model | Df | AIC | BIC | logLik | Deviance | Chisq | Chi Df | p-value |
|-------|-----|--------|--------|--------|----------|-------|--------|---------|
| Null | 5 | −1222.9 | −1195.5 | 616.46 | −1232.9 | | | |
| Full | 8 | −1227.8 | −1184.0 | 621.92 | −1243.8 | 10.91 | 3 | 0.012 |

**Appendix 3—table 2.** Results for the MEGA analysis of the iBEATs data with 3 mo as reference group.

| Term | Estimate | SE | z-value | p-value |
|------|----------|-----|---------|---------|
| Intercept | −1.08 | 0.12 | −8.69 | <0.001 |
| Condition asynchronous | −0.17 | 0.08 | −2.15 | 0.031 |
| 9 mo | 0.57 | 0.16 | 3.48 | <0.001 |
| 18 mo | 0.44 | 0.18 | 2.49 | 0.013 |
| Condition * 9 mo | 0.00 | 0.11 | 0.02 | 0.982 |
| Condition * 18 mo | 0.14 | 0.13 | 1.08 | 0.283 |

Results for the mixed model using a beta error distribution. Results are in comparison to the synchronous condition, and 3 mo age group.

**Appendix 3—table 3.** Results for the MEGA analysis of the iBEATs data with 9 mo as reference group.

| Term | Estimate | SE | z-value | p-value |
|------|----------|-----|---------|---------|
| Intercept | −0.50 | 0.11 | −4.66 | <0.001 |
| Condition asynchronous | −0.17 | 0.07 | −2.49 | 0.013 |
| 3 mo | −0.57 | 0.16 | −3.48 | <0.001 |
| 18 mo | −0.13 | 0.10 | −1.22 | 0.221 |
| Condition * 3 mo | −0.00 | 0.11 | −0.02 | 0.982 |
| Condition * 18 mo | 0.14 | 0.12 | 1.12 | 0.264 |

Results for the mixed model using a beta error distribution. Results are in comparison to the synchronous condition, and 9 mo age group.

**Appendix 3—table 4.** Results for the MEGA analysis of the iBEATs data with 18 mo as reference group.

| Term | Estimate | SE | z-value | p-value |
|------|----------|-----|---------|---------|
| Intercept | −0.63 | 0.13 | −4.91 | <0.001 |
| Condition asynchronous | −0.03 | 0.10 | −0.31 | 0.756 |
| 3 mo | 0.44 | 0.18 | 2.49 | 0.013 |
| 9 mo | −0.13 | 0.10 | −1.22 | 0.221 |
| Condition * 3 mo | −0.14 | 0.13 | −1.08 | 0.283 |

*Appendix 3—table 4 Continued on next page*

*Appendix 3—table 4 Continued*

| Term | Estimate | SE | z-value | p-value |
|------|----------|-----|---------|---------|
| Condition * 9 mo | –0.14 | 0.12 | –1.12 | 0.264 |

Results for the mixed model using a beta error distribution. Results are in comparison to the synchronous condition, and 18 mo age group.

### iBREATH

Next, we report the same results for the iBREATH MEGA-analysis. *Table 5* reports the full-null model comparison, while *Table 6* reports coefficients with 3 mo as reference, *Table 7* with 9 mo as reference, and *Appendix 3—table 8* with 18 mo as reference.

**Appendix 3—table 5.** Full-null model comparison for the iBREATH model.

| Model | Df | AIC | BIC | logLik | Deviance | Chisq | Chi Df | p-value |
|-------|-----|-------|--------|--------|----------|-------|--------|---------|
| Null | 5 | –1600.3 | –1574.5 | 805.14 | –1610.3 | | | |
| Full | 8 | –1600.7 | –1559.4 | 808.37 | –1616.7 | 6.45 | 3 | 0.091 |

**Appendix 3—table 6.** Results for the MEGA analysis of the iBREATH data.

| Term | Estimate | SE | z-value | p-value |
|------|----------|-----|---------|---------|
| Intercept | –1.35 | 0.13 | –10.35 | <0.001 |
| Condition asynchronous | –0.15 | 0.09 | –1.74 | 0.082 |
| 9 mo | 0.25 | 0.17 | 1.47 | 0.141 |
| 18 mo | –0.15 | 0.19 | –0.77 | 0.440 |
| Condition * 9 mo | 0.02 | 0.12 | 0.17 | 0.864 |
| Condition * 18 mo | 0.23 | 0.16 | 1.50 | 0.134 |

Results for the mixed model using a beta error distribution. Results are in comparison to the synchronous condition, and 3 mo age group.

**Appendix 3—table 7.** Results for the MEGA analysis of the iBREATH data.

| Term | Estimate | SE | z-value | p-value |
|------|----------|-----|---------|---------|
| Intercept | –1.10 | 0.11 | –9.76 | <0.001 |
| Condition asynchronous | –0.13 | 0.08 | –1.75 | 0.080 |
| 3 mo | –0.25 | 0.17 | –1.47 | 0.141 |
| 18 mo | –0.40 | 0.13 | –3.12 | 0.001 |
| Condition * 3 mo | –0.02 | 0.12 | –0.17 | 0.864 |
| Condition * 18 mo | 0.21 | 0.15 | 1.43 | 0.154 |

Results for the mixed model using a beta error distribution. Results are in comparison to the synchronous condition, and 9 mo age group.

**Appendix 3—table 8.** Results for the MEGA analysis of the iBREATH data.

| Term | Estimate | SE | z-value | p-value |
|------|----------|-----|---------|---------|
| Intercept | –1.35 | 0.13 | –10.35 | <0.001 |
| Condition asynchronous | –0.15 | 0.09 | –1.74 | 0.082 |
| 3 mo | 0.25 | 0.17 | 1.47 | 0.141 |
| 9 mos | –0.15 | 0.19 | –0.77 | 0.440 |
| Condition * 3 mo | –0.23 | 0.16 | –1.50 | 0.134 |

*Appendix 3—table 8 Continued on next page*

*Appendix 3—table 8 Continued*

| Term | Estimate | SE | z-value | p-value |
|------|----------|-----|---------|---------|
| Condition * 9 mo | −0.21 | 0.15 | −1.43 | 0.154 |

Results for the mixed model using a beta error distribution. Results are in comparison to the synchronous condition, and 18 mo age group.

## Relationship between iBEATs and iBREATH

Regarding the relationship between iBEATs and iBREATH we conducted a beta regression with iBEATH as outcome, and iBEATs, age, as well as the interaction between iBEATs and age as predictors. Here, we report detailed results for each age group (*Appendix 3—table 9*: 3 mo, *Appendix 3—table 10*: 9 mo, *Appendix 3—table 11*: 18 mo).

**Appendix 3—table 9.** MEGA analysis for the relationship between iBEATs and iBREATH with 3 mo as reference group.

| Term | Estimate | SE | z-value | p-value |
|------|----------|-----|---------|---------|
| Intercept | −1.17 | 0.22 | −5.25 | <0.001 |
| iBEATs score | −1.83 | 0.97 | −1.89 | 0.059 |
| 9 mo | −0.15 | 0.34 | −0.42 | 0.674 |
| 18 mo | −0.05 | 0.35 | −0.15 | 0.880 |
| iBEATs * 9 mo | 0.67 | 1.31 | 0.51 | 0.610 |
| iBEATs * 18 mo | 3.13 | 1.41 | 2.22 | 0.027 |

Results for the mixed model using a beta error distribution. Results are in comparison to the 3 mo age group.

**Appendix 3—table 10.** MEGA analysis for the relationship between iBEATs and iBREATH with 9 mo as reference group.

| Term | Estimate | SE | z-value | p-value |
|------|----------|-----|---------|---------|
| Intercept | −1.13 | 0.27 | −4.89 | <0.001 |
| iBEATs score | −1.16 | 0.90 | −1.30 | 0.192 |
| 3 mo | 0.15 | 0.34 | 0.42 | 0.674 |
| 18 mo | 0.09 | 0.38 | 0.24 | 0.810 |
| iBEATs * 3 mo | −0.67 | 1.31 | −0.51 | 0.610 |
| iBEATs * 18 mo | 2.45 | 1.36 | 1.81 | 0.070 |

Results for the mixed model using a beta error distribution. Results are in comparison to the 9 mo age group.

**Appendix 3—table 11.** MEGA analysis for the relationship between iBEATs and iBREATH with 18 mo as reference group.

| Term | Estimate | SE | z-value | p-value |
|------|----------|-----|---------|---------|
| Intercept | −1.22 | 0.28 | −4.39 | <.001 |
| iBEATs score | 1.30 | 1.02 | 1.27 | .204 |
| 3 mo | 0.05 | 0.35 | 0.15 | .880 |
| 9 mos | −0.09 | 0.38 | −0.24 | .810 |
| iBEATs * 3 mo | −3.13 | 1.41 | −2.22 | .027 |
| iBEATs * 9 mo | −2.45 | 1.36 | −1.81 | .070 |

Results for the mixed model using a beta error distribution. Results are in comparison to the 18 mo age group.

## Appendix 4

### Specification curve analysis

For the specification curve analysis, we followed the approach outlined in *Simonsohn et al., 2020*. First, we identified the subset of suitable analytical choices by reviewing all available papers that used a task similar to the iBEATs published so far (*Charbonneau et al., 2022*; *Maister et al., 2017*; *Weijs et al., 2023*). Building up, we extracted potential analytical decisions applicable to our dataset (*Tables 1 and 2*). Second, we ran all suitable analyses and plotted the results (*Figures 1 and 2*). Third, we used a permutation approach to investigate how inconsistent the obtained results were with the null hypothesis of no effect (*Table 3*). Next, we will discuss results of the specification curve analysis for the iBEATs and the iBREATH, respectively, and make recommendations for future data analysis for projects using iBEATs/iBREATH as well as preferential looking paradigms in general.

### Specification curve analysis iBEATs

An overview over analytical choices can be found in *Table 1*. For the iBEATs we identified 7 different categories, yielding a total of 1024 potential analyses (*Figure 2*). We found that 458 (44.73%) analyses led to a significant effect. Most of these (442, 43.16%) yielded a significant effect for a mean synchronicity preference. However, there are also a few analysis paths (16, 1.56%) that we could have chosen that would have resulted in a mean preference for asynchronous stimuli.

**Appendix 4—table 1.** Analytical decisions for the iBEATs.

| Category | Implementation specification curve analysis | Number of analytical choices |
|---|---|---|
| Outlier rejection | Only for sync trials, apply to all trials | 2 |
| SD outlier rejection | 2SD, 2.5SD, 3SD, no criterion | 4 |
| ECG artifact trial rejection | All R-peaks identified, missed single R-peaks, missed R-peaks in last two seconds, identified at least 85% of R-peaks | 4 |
| Data transformation | Log-transformation, non-transformed data | 2 |
| Trial removal | Remove trials with 0 looking times, keep trials with 0 looking times | 2 |
| Min. number of trials per id to be included | 8, 4, 2, 0 | 4 |
| Statistical analysis | Paired t-test, linear mixed model (trials clustered in id) | 2 |

### Specification curve analysis iBREATH

As this is the first paper on respiratory interoceptive sensitivity in infants, for the iBREATH we adapted the choices made for the iBEATs paradigm. An overview can be found in *Table 2*. The only difference between the iBEATs and the iBREATH specification curve analysis concerns artifact removal for the physiological data. For the iBEATs, there were four different categories, while for the iBREATH we identified 6 different categories. There were 1536 potential analyses for the iBREATH (*Figure 3*). We found that 269 (17.51%) analytical choices led to a significant effect. Furthermore, all analyses rendering a significant effect revealed a mean synchronous preference.

**Appendix 4—table 2.** Analytical decisions for the iBREATH.

| Category | Implementation specification curve analysis | Number of analytical choices |
|---|---|---|
| Outlier rejection | Only for sync trials, apply to all trials | 2 |
| SD outlier rejection | 2SD, 2.5SD, 3SD, no criterion | 4 |

*Appendix 4—table 2 Continued on next page*

*Appendix 4—table 2 Continued*

| Category | Implementation specification curve analysis | Number of analytical choices |
|---|---|---|
| Respiration artifact trial rejection | Only good signals, include deep breaths, artifacts in ECG but not respiration, deep breaths & artifacts in ECG, short flat lines in respiratory signal, flat lines & deep breaths and/or ECG artifacts | 6 |
| Data transformation | Log-transformation, non-transformed data | 2 |
| Trial removal | Remove trials with 0 looking times, keep trials with 0 looking times | 2 |
| Min. number of trials per id to be included | 8, 4, 2, 0 | 4 |
| Statistical analysis | Paired t-test, linear mixed model (trials clustered in id) | 2 |

**Appendix 4—table 3.** Inference of the specification curve analysis.

| Test statistic used | Observed result | p-value (% of shuffled results as or more extreme than observed results) |
|---|---|---|
| **iBEATs** | | |
| 1. Median effect size | 0.111 | 0.008 |
| 2. Share of significant results | 470 | 0.042 |
| 3. Aggregate all p-values | Stouffer Z=−39.91 | 0.068 |
| **iBREATH** | | |
| 1. Median effect size | 0.109 | 0.032 |
| 2. Share of significant results | 269 | 0.088 |
| 3. Aggregate all p-values | Stouffer Z=−24.32 | 0.120 |

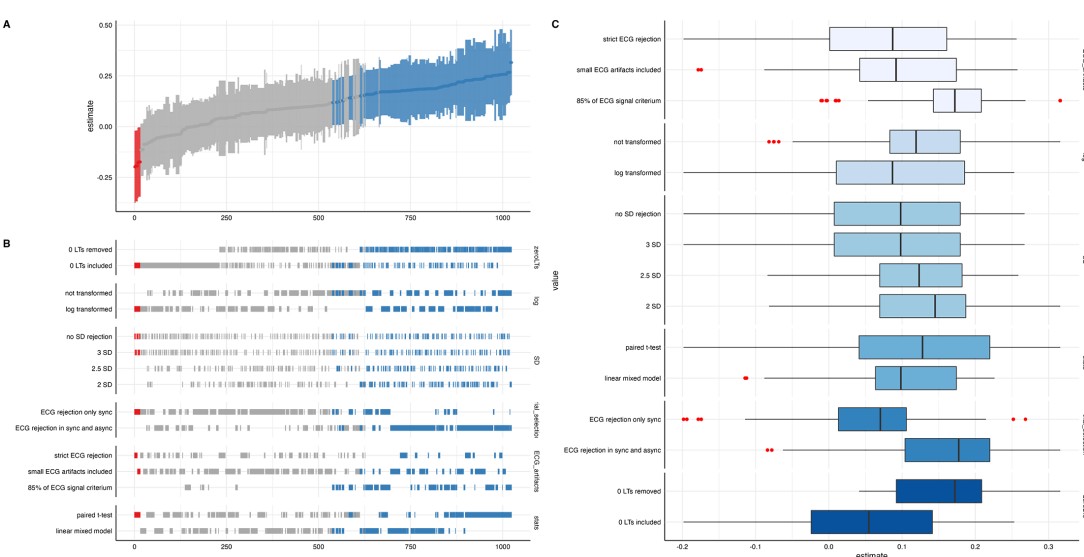

**Appendix 4—figure 1.** Descriptive results from the specification curve analysis for the iBEATs task. Blue coloring in (**A**) and (**B**) refers to a significant result for a mean synchronous preference, while red color indicates to a significant result for a mean asynchronous preference (p<0.05) for the specification and test. In (**A**) standardized beta regression estimates are plotted. In (**B**) an overview for a range of analytical choices is given. In (**C**) analytical choices are further decomposed.

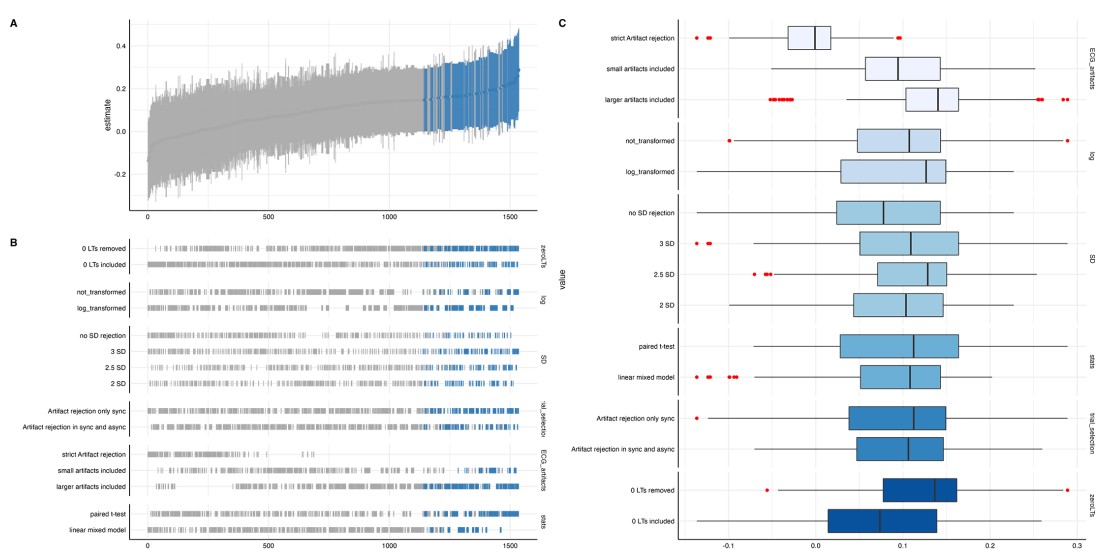

**Appendix 4—figure 2.** Descriptive results from the specification curve analysis for the iBREATH task. Blue coloring in (**A**) and (**B**) refers to a significant result for a mean synchronous preference, while red color indicates to a significant result for a mean asynchronous preference ($p < 0.05$) for the specification and test. In (**A**) standardized beta regression estimates are plotted. In (**B**) an overview for a range of analytical choices is given. In (**C**) analytical choices are further decomposed.

# Appendix 5

## Slow and fast asynchronous trials

Asynchronous trials in the iBEATs and the iBREATH could be either faster or slower than the infant's respective physiological signal. In an exploratory analysis, we computed t-tests to investigate whether looking times differed between slow and fast asynchronous trials for all age groups and paradigms (*Table 1*). For this analysis we used the pre-processing of the MEGA-analysis. Thus, looking times might differ from the confirmatory analysis, as different criteria regarding e.g., regarding outlier rejection were used. Overall, we did not find evidence for a difference between looking times to fast and slow asynchronous trials.

**Appendix 5—table 1.** Looking times for slow and fast asynchronous trials.

|  | *Mean* Async (*SD*) in ms | *Mean* Slow (*SD*) in ms | *Mean* Fast (*SD*) in ms | Comparison |
|---|---|---|---|---|
| iBEATs, 3 mo | 5812 (5804) | 5652 (5731) | 5961 (5885) | t(299.15)=0.46, p=0.64 |
| iBEATs, 9 mo | 6539 (4924) | 6517 (5125) | 6655 (4755) | t(536.77)=0.33, p=0.74 |
| iBEATs, 18 mo | 6877 (5518) | 6608 (5473) | 7120 (5578) | t(171.77)=0.61, p=0.54 |
| iBREATH, 3 mo | 7259 (8057) | 7924 (8601) | 6630 (7491) | t(206.34)=−1.18, p=0.24 |
| iBREATH, 9 mo | 6123 (5282) | 6657 (5779) | 5659 (4939) | t(366.26)=−1.85, p=0.066 |
| iBREATH, 18 mo | 5477 (5325) | 6078 (6053) | 4969 (4610) | t(99.74)=−1.11, p=0.27 |

For this analysis we used the pre-processing of the MEGA-analysis. Thus, looking times might differ from the confirmatory analysis.

