## [Editor Report · eLife Assessment]

This study presents **important** findings on the early development of cardiac and respiratory interoceptive sensitivity based on an investigation of infants aged 3, 9 and 18 months and on extensive statistical analyses. The evidence supporting the conclusions are **convincing** although the research faced technical challenges that limited part of the findings interpretation. This study will be of significant interest to developmental psychologists and neuroscientists working on interoception and its influence on socio-cognitive development.

---

## [Referee Report · Reviewer #1 (Public review)]

Summary:

The authors of this study investigated the development of interoceptive sensitivity in the context of cardiac and respiratory interoception in 3-, 9-, and 18-month-old infants using a combination of both cross-sectional and longitudinal designs. They utilised the cardiac interoception paradigm developed by Maister et al (2017) and also developed a new paradigm to investigate respiratory interoception in infants. The main findings of this research are that 9-month-old infants displayed a preference for stimuli presented synchronously with their own heartbeat and respiration. The authors found less reliable effects in the 18-month-old group, and this was especially true for the respiratory interoceptive data. The authors replicated a visual preference for synchrony over asynchrony for the cardiac domain in 3-month-old infants, while they found inconclusive evidence regarding the respiratory domain. Considering the developmental nature of the study, the authors also investigated the presence of developmental trajectories and associations between the two interoceptive domains. They found evidence for a relationship between cardiac and respiratory interoceptive sensitivity at 18 months only and preliminary evidence for an increase in respiratory interoception between 9 and 18 months.

Strengths:

The conclusions of this paper are mostly well supported by data, and the data analysis procedures are rigorous and well-justified. The main strengths of the paper are:

- A first attempt to explore the association between two different interoceptive domains. How different organ-specific axes of interoception relate to each other is still open and exploring this from a developmental lens can help shed light into possible relationships. The authors have to be commended for developing a novel interoceptive tasks aimed at assessing respiratory interoceptive sensitivity in infants and toddlers, and for trying to assess the relationship between cardiac and respiratory interoception across developmental time.

- A thorough justification of the developmental ages selected for the study. The authors provide a rationale behind their choice to examine interoceptive sensitivity at 3, 9, and 18-months of age. These are well justified based on the literature pertaining to self- and social development. Sometimes, I wondered whether explaining the link between these self and social processes and interoception would have been beneficial as a reader not familiar with the topics may miss the point.

- An explanation of direction of looking behaviour using latent curve analysis. I found this additional analysis extremely helpful in providing a better understanding of the data based on previous research and analytical choices. As the authors explain in the manuscript, it is often difficult to interpret the direction of infant looking behaviour as novelty and familiarity preferences can also be driven by hidden confounders (e.g. task difficulty). The authors provide compelling evidence that analytical choices can explain some of these effects. Beyond the field of interoception, these findings will be relevant to development psychologists and will inform future studies using looking time as a measure of infants' ability to discriminate among stimuli.

- The use of simulation analysis to account for small sample size. The authors acknowledge that some of the effects reported in their study could be explained by a small sample size (i.e. the 3-month-olds and 18-month-olds data). Using a simulation approach, the authors try to overcome some of these limitations and provide convincing evidence of interoceptive abilities in infancy and toddlerhood (but see also my next point).

Comments on revision:

The authors have clearly addressed the comments on the previous version of this manuscript. I have no further comments.

---

## [Referee Report · Reviewer #2 (Public review)]

Summary:

This study by Tünte et al. investigated the development of interoceptive sensitivity during the first year of life, focusing specifically on cardiac and respiratory sensitivity in infants aged 3, 9, and 18 months. The research employed a previously developed experimental paradigm for the cardiac domain and adapted it for a novel paradigm in the respiratory domain. This approach assessed infants' cardiac and respiratory sensitivity based on their preferential looking behavior toward visuo-auditory stimuli displayed on a monitor, which moved either in sync or out of sync with the infants' own heartbeats or breathing. The results in the cardiac domain showed that infants across all age groups preferred stimuli moving synchronously rather than asynchronously with their heartbeat, suggesting the presence of cardiac sensitivity as early as 3 months of age. However, it is noteworthy that this preference direction contradicts a previous study, which found that 5-month-old infants looked longer at stimuli moving asynchronously with their heartbeat (Maister et al., 2017). In the respiratory domain, only the group of 9-month-old infants showed a preference for stimuli presented synchronously with their breathing. The authors conducted various statistical analyses to thoroughly examine the obtained data, providing deeper insights valuable for future research in this field.

Strengths:

Few studies have explored the early development of interoception, making the replication of the original study by Maister et al. (2017) particularly valuable. Beyond replication, this study expands the investigation into the respiratory domain, significantly enhancing our understanding of interoceptive development. The provision of longitudinal and cross-sectional data from infants at 3, 9, and 18 months of age is instrumental in understanding their developmental trajectory.

Weaknesses:

Due to a technical error, this study failed to counterbalance the conditions of the first trial in both the iBEAT and iBREATH tests. Although the authors addressed this issue as much as possible by employing alternative analyses, it should be noted that this error may have critically influenced the results and, thus, the conclusions.

---

## [Author Response]

The following is the authors’ response to the previous reviews.

We sincerely appreciate the time and effort you and the reviewers have invested in evaluating our work.

We are grateful for the constructive criticism of the reviewers. Building up on their feedback we have made additions to the reviewed preprint. Specifically, we have added information to the supplementary materials to give additional context on the impact of the fixed experimental design on infants’ looking behavior. Further, we have adapted the text throughout the manuscript to incorporate a thorough discussion of the impact of the experimental design.

We believe that these revisions and the inclusion of supplementary analyses provide a clearer understanding of our findings.